# The microbiota promotes social behavior by modulating microglial remodeling of forebrain neurons

**Joseph J. Bruckner**[1], **Sarah J. Stednitz**[1¤], **Max Z. Grice**[1], **Dana Zaidan**[1], **Michelle S. Massaquoi**[2], **Johannes Larsch**[3], **Alexandra Tallafuss**[1], **Karen Guillemin**[2,4], **Philip Washbourne**[1]*, **Judith S. Eisen**[1]*

**1** Institute of Neuroscience, Department of Biology, University of Oregon, Eugene, Oregon, United States of America, **2** Institute of Molecular Biology, Department of Biology, University of Oregon, Eugene, Oregon, United States of America, **3** Department Genes-Circuits-Behavior, Max Planck Institute of Neurobiology, Martinsried, Germany, **4** Humans and the Microbiome Program, CIFAR, Toronto, Ontario, Canada

¤ Current address: Queensland Brain Institute, University of Queensland, St. Lucia, Queensland, Australia
* pwash@uoregon.edu (PW); eisen@uoregon.edu (JSE)

**Data Availability Statement:** Single-cell RNA sequencing data are presented within the paper and its Supporting Information files. Individual numerical source data for all other experiments are

## Abstract

Host-associated microbiotas guide the trajectory of developmental programs, and altered microbiota composition is linked to neurodevelopmental conditions such as autism spectrum disorder. Recent work suggests that microbiotas modulate behavioral phenotypes associated with these disorders. We discovered that the zebrafish microbiota is required for normal social behavior and reveal a molecular pathway linking the microbiota, microglial remodeling of neural circuits, and social behavior in this experimentally tractable model vertebrate. Examining neuronal correlates of behavior, we found that the microbiota restrains neurite complexity and targeting of forebrain neurons required for normal social behavior and is necessary for localization of forebrain microglia, brain-resident phagocytes that remodel neuronal arbors. The microbiota also influences microglial molecular functions, including promoting expression of the complement signaling pathway and the synaptic remodeling factor *c1q*. Several distinct bacterial taxa are individually sufficient for normal microglial and neuronal phenotypes, suggesting that host neuroimmune development is sensitive to a feature common among many bacteria. Our results demonstrate that the microbiota influences zebrafish social behavior by stimulating microglial remodeling of forebrain circuits during early neurodevelopment and suggest pathways for new interventions in multiple neurodevelopmental disorders.

## Introduction

Impaired social behavior is a hallmark of multiple neurodevelopmental disorders, including autism spectrum disorder (ASD) and schizophrenia [1]. However, the organization, function, and development of brain circuits underlying social interactions are poorly understood, and effective interventions in these disorders are elusive. Altered intestinal microbiota composition

available via figshare (https://figshare.com/projects/Bruckner_et_al_Data/136756). The scRNAseq data has been uploaded to NCBI BioSample under the following accession numbers: BioProject ID (PRJNA885906) BioSample IDs (SAMN31112120 and SAMN31112122).

**Funding:** Life Sciences Research Foundation (https://lsrf.org) award to J.J.B.; National Institute of Mental Health (https://www.nimh.nih.gov) award F32MH118809 to J.J.B.; Eunice Kennedy Shriver National Institute of Child Health and Human Development (https://www.nichd.nih.gov) award T32HD007348 to S.J.S.; Oregon Developmental Biology Collaborations (https://ohsu-uopartnership.uoregon.edu) award to S.J.S.; Eunice Kennedy Shriver National Institute of Child Health and Human Development (https://www.nichd.nih.gov) award R25HD070817 to M.Z.G.; National Science Foundation Bio/DBI (https://www.nsf.gov) award BIO/DBI1758015 to D.Z.; Eunice Kennedy Shriver National Institute of Child Health and Human Development (https://www.nichd.nih.gov) award T32HD007348 to M.S.M.; National Institute of Mental Health (https://www.nimh.nih.gov) award R21MH104188 to J.S.E.; National Institute of Mental Health (https://www.nimh.nih.gov) award R21MH104188 to P.W.; National Institute of Mental Health (https://www.nimh.nih.gov) award R33MH104188 to J.S.E.; National Institute of Mental Health (https://www.nimh.nih.gov) award R33MH104188 to P.W.; John Simon Guggenheim Memorial Foundation (https://www.gf.org) award to J.S.E.; National Institute of General Medical Sciences (https://www.nigms.nih.gov) award P01GM125576 to J.S.E.; National Institute of General Medical Sciences (https://www.nigms.nih.gov) award P01GM125576 to K.G.; Gordon and Betty Moore Foundation (https://www.moore.org) Symbiosis Investigator Award GBMF9205 (https://doi.org/10.37807/GBMF9205) to J.S.E. The funders had no role in study design, data collection and analysis, decision to publish, or preparation of the manuscript.

**Competing interests:** The authors have declared that no competing interests exist.

**Abbreviations:** ANOVA, one-way analysis of variance; ASD, autism spectrum disorder; BDNF, brain-derived neurotrophic factor; CMTK, Computational Morphology Toolkit; CVZ, conventionalized; dpf, days post fertilization; EM, embryo medium; GF, germ-free; GO, gene ontology; hpf, hours post fertilization; Itgax, integrin subunit alpha X; NAM, neurogenic associated microglia; PBS, phosphate-buffered saline; PBSTx, phosphate-buffered saline with 0.5% Triton X-100; SAM, synaptic region–

is also associated with multiple neurodevelopmental disorders, yet how the microbiota contributes to human health is still obscure [2]. Though several clinical studies show promising improvement in ASD behavioral symptoms following gut–brain axis interventions, optimal parameters for these interventions and the underlying mechanisms remain unclear [3].

Zebrafish are increasingly employed to explore the microbiota–gut–brain axis and are an excellent model for understanding how the microbiota influences development of the "social brain" and generating insights to inform interventions in humans [4]. Development of the early circuitry that regulates mammalian social behavior is difficult to observe in the prenatal brain, whereas equivalent neurodevelopment is readily visualized in vivo in transparent larval zebrafish. Zebrafish are naturally gregarious and manifest social traits including shoaling, aggression, kin recognition, and orienting as early as 12 to 16 days post fertilization (dpf) [5–10]. Combining the genetic and experimental accessibility of zebrafish, we can identify precise developmental events that facilitate normal social behavior and that may go awry in neurodevelopmental disorders.

Although the entire circuitry for social interactions is still under investigation, we and others have shown that zebrafish ventral nuclei of the area ventralis telencephali (Vv) are required for normal social behavior [6,11,12]. Connectomic studies suggest that Vv may integrate input from the midbrain and olfactory bulb and send efferent projections to higher-order processing centers in the habenula, hypothalamus, and preoptic area [13–16]. Normal Vv circuit connectivity is likely established long before social orienting is expressed at 14 dpf, so the rapid, sequential development of social characteristics could represent ongoing refinement of cells required to execute social behavior [5]. For example, development of many neuronal circuits is characterized by a critical period of initial outgrowth and synapse formation followed by pruning of superfluous connections and strengthening of specific nodes [17]. Our previous work identified a subpopulation of Vv neurons required for normal zebrafish social orienting and place preference, which, for simplicity, we refer to according to the Gal4 enhancer trap transgene that labels them, y321 (vTel$^{y321}$) [6,18]. Understanding how intrinsic and extrinsic factors influence vTel$^{y321}$ neuronal arbor refinement during early critical periods will enable us to predict features that can modify behavioral deficits in social disorders.

Microglia, the brain's resident myeloid cells, have well-defined roles regulating brain development and function, including modifying neuronal morphology by directing axon outgrowth and refining synapses [19]. Microglia are also required in the early postnatal brain for the development of normal social behavior [20–24]. Microglia infiltrate the brain in multiple waves [25,26]. In zebrafish, this process begins with primitive microglia that differentiate in the rostral blood island, infiltrate the brain around 2.5 dpf, and persist through larval stages [27,28]. These microglia are eventually replaced by a second population that differentiates from hematopoietic stem cells in the dorsal aorta and infiltrates the brain beginning at approximately 14 dpf [26,29]. In the larval brain, microglia actively survey the surrounding tissue, modulate neuronal activity, and phagocytose neuronal material [30,31]. How microglia remodel neural circuits remains an active area of investigation. The classical complement cascade is one of the best-understood pathways employed by microglia to sculpt neural circuits [32]. For example, complement component C1q is thought to tag axons and synapses, initiating the complement cascade and subsequent synaptic pruning events.

It is increasingly appreciated that host-associated microbes can shape social behavior by influencing neurodevelopment [22,33,34]. Mice raised germ-free (GF) or with an abnormal microbiota exhibit impaired social behavior, which is correlated with microbial modulation of neuronal gene expression, neurotransmitter levels, brain maturation, and myelination [35–41]. Host-associated microbes influence social behavior across taxa. For example, the *Drosophila melanogaster* microbiota promotes social preference through serotonergic signaling [42].

associated microglia; scRNAseq, single-cell RNA
sequencing; SPF, specific pathogen-free; SSC,
saline sodium citrate; Vv, ventral nuclei of the area
ventralis telencephali; XGF, ex germ-free.

GF zebrafish have abnormal anxiety-related and locomotor behaviors that can be attenuated by probiotic administration, an intervention that also influences shoaling behavior via *brain-derived neurotrophic factor* (*BDNF*) and serotonin signaling [43–45]. However, these probiotic *Lactobacillus* strains were applied to adult zebrafish and do not normally populate the zebrafish intestine, so it is unclear whether microbial modulation occurs through a similar mechanism during normal neurodevelopment of circuits that regulate social behavior [46,47]. Like many circulating immune cells, microglia are responsive to microbial signals, and the microbiota appears to influence normal microglial colonization, maturation, morphology, activation, and homeostasis [22,48,49]. However, how microbial modulation of microglial function feeds forward to influence neural circuit architecture, especially in brain regions that regulate social behavior, has not been studied.

In this study, we use gnotobiotic techniques to identify a neuroimmune pathway linking the microbiota and a region of the zebrafish brain that regulates social behavior. First, we found that normal social behavior in late-flexion larvae (around 14 dpf) requires microbes earlier in development, suggesting a critical period for microbial input. We then reconstructed hundreds of individual vTel$^{y321}$ neurons and observed significant microbial modulation of arbor complexity and targeting during the period when social behavior is developing. We found that microglia are critical for remodeling forebrain neurites during early circuit development, so we tested the hypothesis that the microbiota influences zebrafish forebrain microglia. We discovered that the microbiota promotes forebrain microglial abundance and gene expression, including promoting expression of complement pathway genes involved in arbor remodeling. We found that diverse strains of zebrafish-associated bacteria are capable of restoring neuronal and microglial phenotypes in GF fish. Together, our experiments suggest that a feature common to many bacterial taxa promotes social behavior by stimulating host innate immune pathways that redistribute forebrain microglia, alter microglial function, and remodel neuronal arbors.

## Results

### The microbiota promotes zebrafish social behavior

Social orienting among pairs of zebrafish is robust at approximately 14 dpf [5], suggesting that neuronal circuits that facilitate this behavior develop much earlier. Microbial factors, such as those from bacteria that colonize the gut once it is patent at approximately 4 dpf, could influence these neurodevelopmental events (Fig 1A) [50]. To test the hypothesis that normal social behavior development specifically requires the microbiota early, before social behavior is expressed, we raised zebrafish GF for the first week of life, inoculated them with a normal microbiota at 7 dpf, and assessed social behavior at 14 dpf with our previously described social orienting assay (XGF; Fig 1A and 1B) [6,50]. This assay accurately reproduces and measures social orienting behavior exhibited by freely swimming larvae and adults [5,6]. Compared to "conventionalized" siblings derived GF and then inoculated with a normal microbiota on day 0 (CVZ), XGF larvae spend significantly less time than CVZ controls in close proximity to and oriented at 45 to 90˚ to the stimulus fish (Fig 1C–1F, S1 and S2 Movies). These results show that an intact microbiota is required early for later development of normal social behavior.

It is possible that the microbiota does not usually guide social neurodevelopment, but rather that removing the host-associated microbiota for the first week of life causes nutritional deficits that simply delay normal development. Standard length, a commonly used measure of zebrafish development, is slightly but statistically significantly reduced in XGF larvae relative to their CVZ siblings (S1A Fig) [51]. To address the possibility that developmental delay accounts for social deficits of XGF fish, we binned social orienting measurements according to the

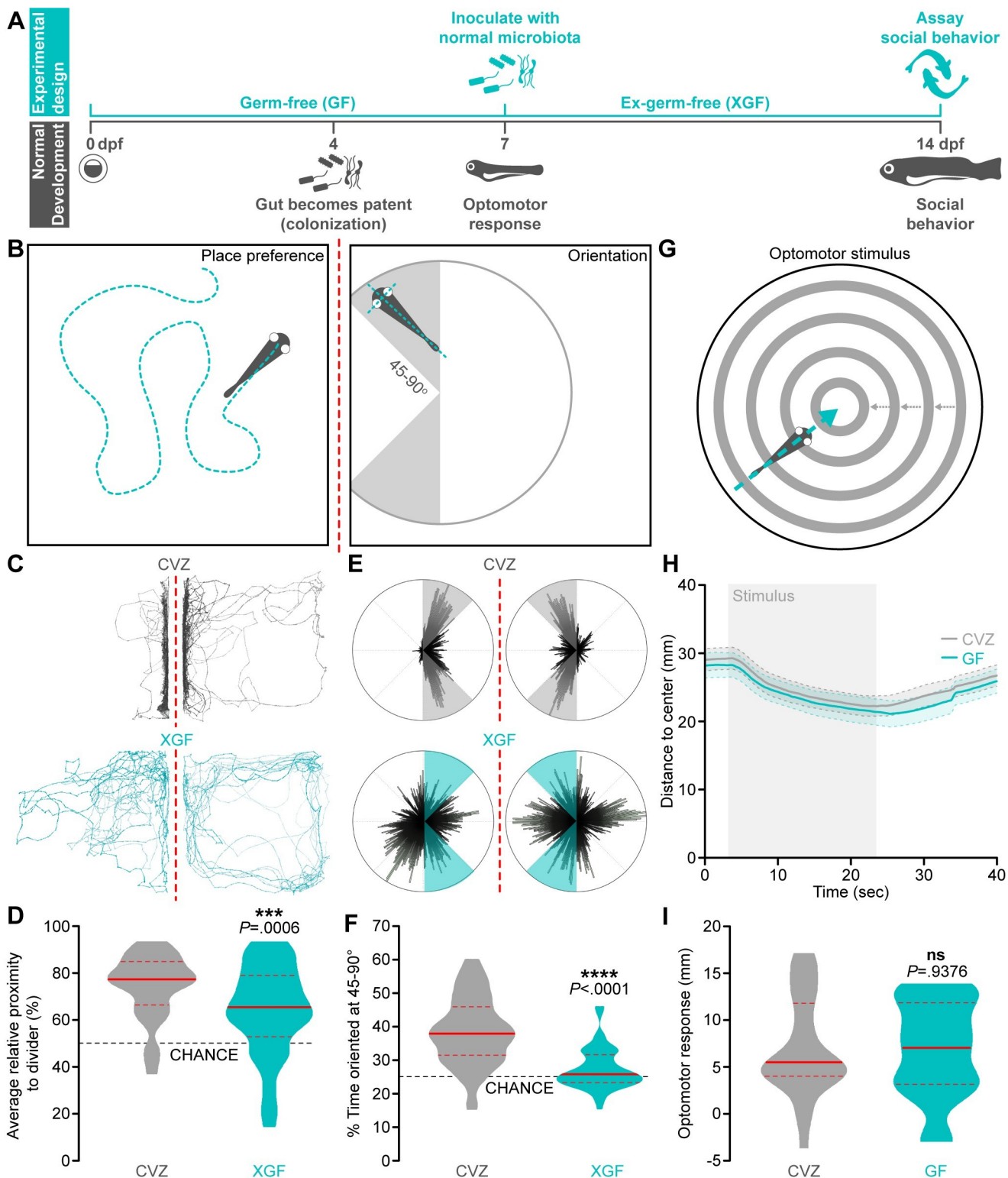

**Fig 1. The microbiota promotes zebrafish social behavior.** (**A**) Experimental design timeline. (**B**) Zebrafish social behavior is assessed by measuring place preference (left) and body orientation (right) in paired fish separated by a transparent divider (dotted red line). (**C-F**) Social behavior is reduced in 14 dpf XGF larvae relative to CVZ siblings. Traces (**C**) and 360° body position polar plots (**E**) of representative CVZ (gray) and XGF (aqua) larvae during social behavior. Average relative proximity to the transparent divider (**D**) and percent of time oriented at 45–90° (**F**) are significantly reduced in XGF (*n* = 67) larvae relative to CVZ controls (*n* = 54 larvae; Mann–Whitney *U* test). (**G**) Sensorimotor integration is assessed by measuring distance traveled in response

to a stimulus simulating motion toward the dish center. (**H**) Average distance to center is similar in 7 dpf CVZ (gray, $n$ = 25) and GF (aqua, $n$ = 20) larvae during and following stimulus presentation (gray bar; solid lines represent mean, dotted lines represent SEM). (**I**) Distance traveled in response to optomotor stimulus is not significantly reduced in GF larvae relative to CVZ siblings (unpaired $t$ test). ns, not significant; ***, $P$ < .001; ****, $P$ < .0001. Solid red line represents the median; dotted red lines represent the upper and lower quartiles. Data underlying this figure are available on figshare: https://figshare.com/projects/Bruckner_et_al_Data/136756.

standard length of each fish. We do not observe a difference in social orienting between XGF larvae and CVZ siblings that are 5 mm or smaller, likely because these stunted fish are unable to execute social orienting (S1B Fig). However, social orienting remains decreased in XGF larvae size matched to CVZ controls that are 6 mm or longer (S1B Fig). Smaller standard length is significantly correlated with reduced orienting behavior ($P$ = .02). However, when treatment condition is considered as a covariate using multiple regression, it is significantly predictive ($P$ = .003) while length is not ($P$ = .144). Therefore, treatment condition and not reduced size primarily accounts for impaired XGF social behavior.

To execute normal social orienting behavior, larval zebrafish must be able to visually detect a conspecific and rapidly change body position to reciprocate movements of the other fish. It is possible that the microbiota influences circuitry underlying the early vision and locomotion required for social behavior. To address this possibility, we simultaneously assayed vision and locomotion by comparing kinetics of the optomotor response to virtual motion in 7 dpf GF larvae and CVZ controls. We presented larvae with a full-field optomotor stimulus composed of concentric rings simulating motion toward the center of an experimental chamber [52]. This stimulus induces fish to swim toward the chamber center, followed by dispersal toward the edge after the stimulus ceases (Fig 1G). We observe no significant differences in the kinetics or magnitude of responses to optomotor stimulus in GF fish relative to their CVZ siblings, suggesting that the microbiota does not influence early development of vision or motor output (Fig 1H and 1I). Additionally, average swim speed remains normal in 14 dpf XGF larvae relative to CVZ controls (S1C Fig), suggesting that the microbiota influences circuits specific to social behavior directly, rather than by modulating vision or locomotion.

## The microbiota restrains vTel$^{y321}$ neuronal arborization

As vTel$^{y321}$ neurons are required for normal social behavior [6], we hypothesized that the microbiota might promote social behavior by modulating the number of vTel$^{y321}$ cells. The vTel$^{y321}$ nucleus comprises an average of 229 neurons in 7 dpf CVZ larvae, which is slightly but significantly reduced in GF larvae (Fig 2A and 2B). However, 14 dpf XGF larvae that cannot execute normal social behavior have essentially the same number of vTel$^{y321}$ neurons as their CVZ siblings, indicating that the microbiota does not influence *y321Et* promoter expression or modulate social behavior by promoting vTel$^{y321}$ neuron proliferation (Fig 2A and 2B). We therefore hypothesized that the microbiota could influence social behavior by modulating connectivity of vTel$^{y321}$ neurons. To test this possibility, we visualized individual vTel$^{y321}$ arbors using a sparse mosaic labeling technique, bloSwitch, that inefficiently recombines UAS-driven fluorescent proteins to generate random sparse labeling of Gal4-expressing cells [53]. We used a semiautomated segmentation approach to image, reconstruct, and quantify mosaic RFP-expressing vTel$^{y321}$ neurons and the GFP-expressing reference population from CVZ and GF siblings in 3D (Fig 2A).

Randomly sampling vTel$^{y321}$ neurons across dozens of 7 dpf larval brains, we observed surprisingly diverse morphologies from short neurites with only a few branches to complex arbors hundreds of microns long (Fig 2A; CVZ: $n$ = 73 neurons from 24 larvae; GF: $n$ = 70 neurons from 25 larvae). Though we occasionally observed vTel$^{y321}$ neurites projecting into adjacent

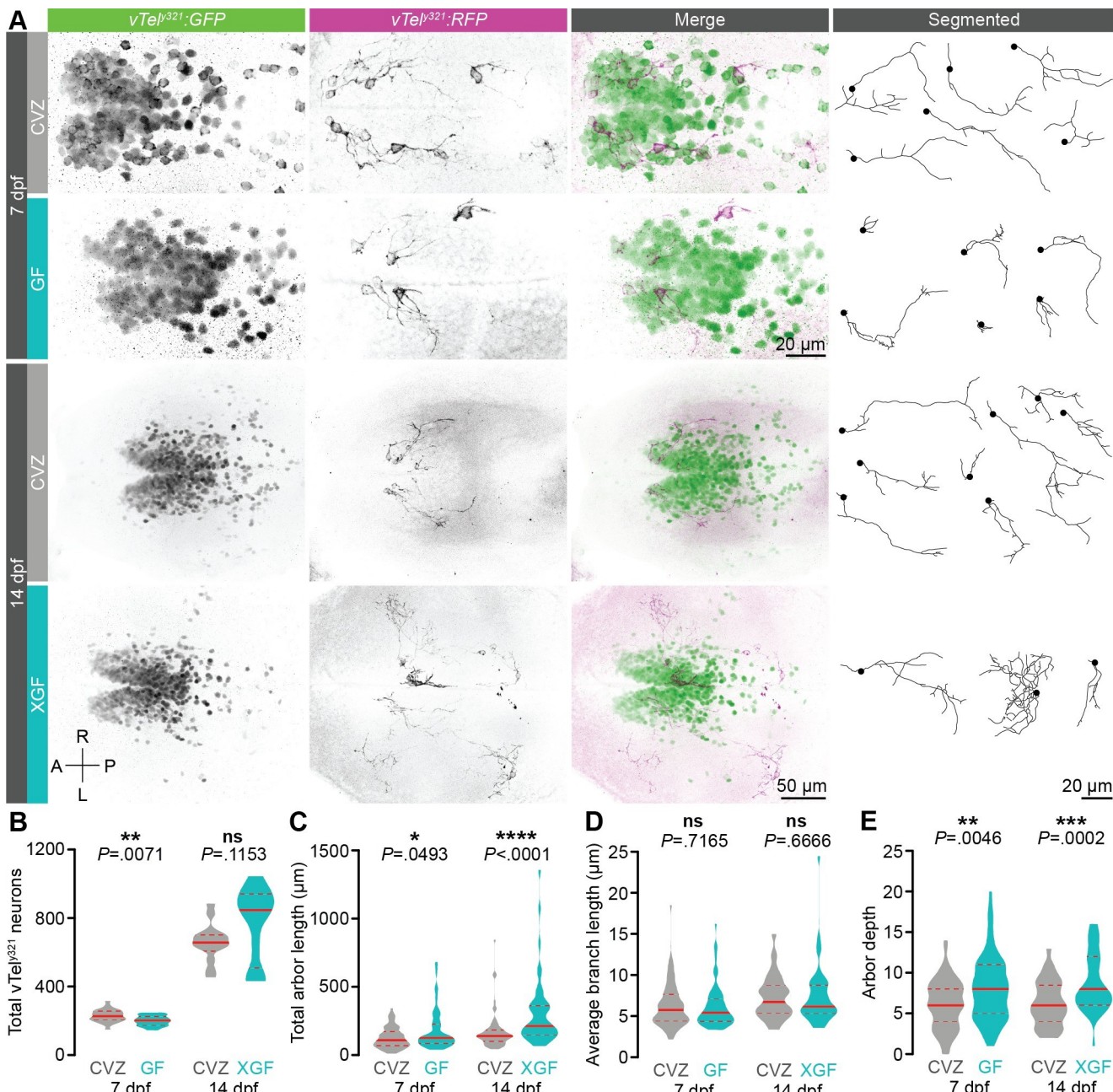

**Fig 2. The microbiota restrains vTel$^{y321}$ arborization.** (**A**) Maximum-intensity projections of vTel$^{y321}$ GFP (green), sparse mosaic vTel$^{y321}$ RFP (magenta), and individually segmented vTel$^{y321}$ neurons from the same representative 7 dpf (top) or 14 dpf (bottom) larvae raised CVZ (gray), GF (aqua), or XGF (aqua). (**B**) The total number of vTel$^{y321}$ GFP neurons is reduced in 7 dpf GF larvae relative to CVZ controls ($n$ = 24 CVZ and 22 GF larvae; unpaired $t$ test), but not in 14 dpf XGF larvae relative to CVZ controls ($n$ = 14 CVZ and 12 XGF larvae; Mann–Whitney $U$ test). (**C-E**) Total arbor length (**C**) and arbor depth (**E**) are increased and average branch length is unchanged (**D**) in 7 dpf GF larvae relative to CVZ controls ($n$ = 73 neurons from 24 CVZ larvae and 69 neurons from 25 GF larvae; Mann–Whitney $U$ tests) and in 14 dpf XGF larvae relative to CVZ controls ($n$ = 69 neurons from 14 CVZ larvae and 46 neurons from 13 XGF larvae; Mann–Whitney $U$ tests). ns, not significant; *, $P < .05$; **, $P < .01$; ***, $P < .001$; ****, $P < .0001$. Solid red line represents the median; dotted red lines represent the upper and lower quartiles. Data underlying this figure are available on figshare: https://figshare.com/projects/Bruckner_et_al_Data/136756.

brain regions including the preoptic area and hypothalamus, fasciculation of these neurites in large tracts made reconstruction of individual arbors impossible and excluded them from analysis. The total length of vTel$^{y321}$ arbors is significantly increased in GF larvae compared to CVZ siblings and the average length of vTel$^{y321}$ branches remains indistinguishable, suggesting that the microbiota restrains vTel$^{y321}$ arbor length by modulating branching rather than outgrowth of individual neurites (Fig 2C and 2D). To assess this possibility, we quantified arbor depth, a measure of branching complexity corresponding to the maximum number of bifurcations on a given arbor. GF vTel$^{y321}$ arbors are significantly deeper than those of CVZ controls, reinforcing the idea that the normal role of the microbiota is to restrain vTel$^{y321}$ neurite branching (Fig 2C and 2D).

To examine whether impaired early vTel$^{y321}$ neurite development persists at later stages when larvae can execute social behavior, we reconstructed and quantified vTel$^{y321}$ arbors in 14 dpf XGF larvae (Figs 1A and 2A; CVZ: $n = 69$ neurons from 14 larvae, XGF: $n = 46$ neurons from 13 larvae). Relative to CVZ siblings, vTel$^{y321}$ arbors from 14 dpf XGF larvae retain the similar average branch length and increased total arbor length and depth observed in 7 dpf larvae (Fig 2C–2E). Since exuberant vTel$^{y321}$ arborization in 7 dpf GF larvae persists to late larval stages of XGF fish that exhibit impaired social behavior despite the reintroduction of the normal microbiota, we conclude that there is an early developmental window during which microbial modulation of neurodevelopment is critical for normal connectivity in circuits required for later expression of social behavior.

## The microbiota guides vTel$^{y321}$ arbor targeting

To assess the spatial organization of vTel$^{y321}$ arbor complexity, we applied 3D Sholl analysis to segmented vTel$^{y321}$ arbors in GF, XGF, and CVZ larvae. Sholl analysis quantifies the number of times each neuronal arbor intersects a series of concentric spheres, or connective zones, centered around the soma and increasing in diameter by 1 μm (insets, Fig 3A and 3B) [54]. vTel$^{y321}$ arbors from 7 dpf CVZ larvae cover a connective zone over 100 μm from the soma, and though arbors from GF larvae cover a similar connective zone, they exhibit a dramatic increase in complexity 10 to 80 μm from the soma (Fig 3A). The total number of Sholl intersections and maximum number of Sholl intersections are not dramatically different in GF larvae relative to CVZ controls; however, a significant increase in both the maximum Sholl radius and Sholl radius that contains the most intersections suggests that the microbiota normally restrains distal arbor complexity (Fig 3C–3F).

As with the measures of arbor complexity described above, an additional week of development following inoculation with a normal microbiota does not restore normal Sholl profiles to XGF vTel$^{y321}$ arbors; in fact, the rearrangement observed at 7 dpf is even more exaggerated. Though 14 dpf CVZ vTel$^{y321}$ arbors can cover a volume up to 160 μm from the soma, XGF vTel$^{y321}$ arbors are nearly twice as complex as those from CVZ controls across the majority of this connective zone (Fig 3B). Total Sholl intersections, maximum Sholl intersections, maximum Sholl radius, and the Sholl radius containing the most intersections are all significantly increased in 14 dpf XGF vTel$^{y321}$ arbors relative to those from CVZ siblings. Therefore, microbial modulation of vTel$^{y321}$ arborization during early development is crucial to restrain a connective zone that continues to elaborate arbors as social behavior coalesces.

The hundreds of neurons that comprise the vTel$^{y321}$ nucleus likely include multiple morphological subtypes. We hypothesized that the microbiota might be required for normal development of specific vTel$^{y321}$ neuronal subtypes. To address this possibility, we extracted 13 morphological parameters from segmented vTel$^{y321}$ neurons and used hierarchical clustering (Fig 3G and 3I) and factor analysis (Fig 3H and 3J) to group them according to morphology.

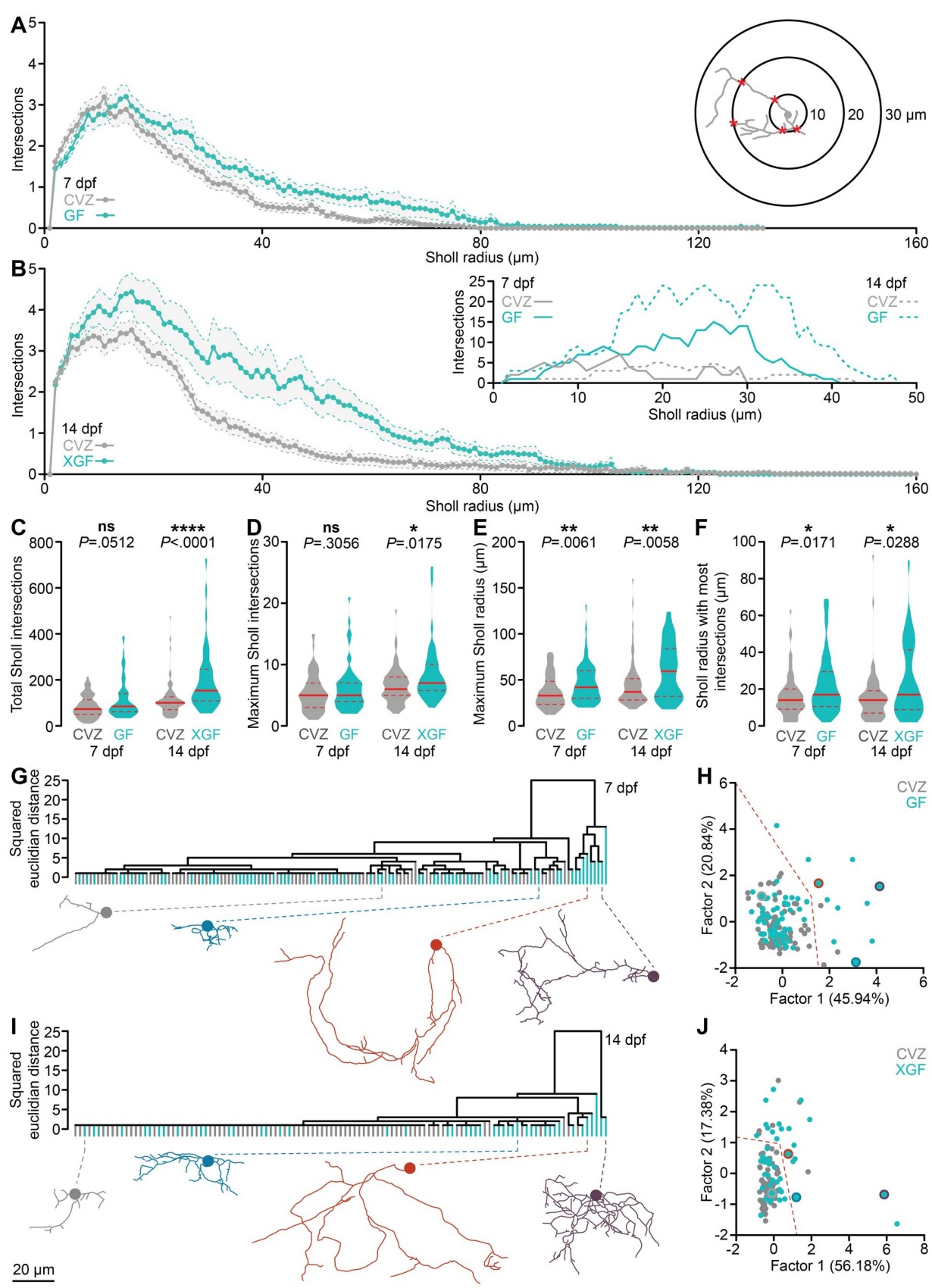

**Fig 3. The microbiota reorganizes vTel$^{y321}$ neurite complexity.** (**A**) Average Sholl profiles (inset) for vTel$^{y321}$ neurons from 7 dpf larvae raised CVZ (gray) or GF (aqua). (**B**) Average Sholl profiles from 14 dpf larvae raised CVZ (gray) or XGF (aqua), and representative examples (inset). (**C-E**) Total Sholl intersections across each arbor (**C**) and maximum Sholl intersections at any radius (**D**) are not different between vTel$^{y321}$ neurons in 7 dpf CVZ and GF larvae, but are increased in vTel$^{y321}$ neurons in 14 dpf XGF larvae relative to CVZ siblings, whereas maximum Sholl radius (**E**) and Sholl radius with the most intersections (**F**) are increased in vTel$^{y321}$ neurons between 7 dpf CVZ and GF larvae and between 14 dpf CVZ and XGF larvae (7 dpf, $n$ = 73 neurons from 24 CVZ larvae, 69 neurons from 25 GF larvae; 14 dpf, $n$ = 69 neurons from 14 CVZ larvae, 46 neurons from 13 XGF larvae; Mann–Whitney $U$ tests). (**G-J**) vTel$^{y321}$ neurons from 7 dpf (**G, H**) larvae raised GF (aqua) or CVZ (gray) or 14 dpf (**I, J**) larvae raised XGF (aqua) or CVZ (gray), grouped by average linkage in hierarchical clustering (**G, I**) or by factor analysis (**H, J**; 7 dpf, $n$ = 73 neurons from 24 CVZ larvae, 69 neurons from 25 GF larvae; 14 dpf, $n$ = 69 neurons from 14 CVZ larvae, 46 neurons from 13 XGF larvae). Representative examples are included below each dendrogram and indicated by color in factor analysis plots. Dotted orange lines in (**H**) and (**J**) roughly delineate complex and simple neuronal morphologies, which are colored orange in S3 Fig. ns, not significant; *, $P < .05$; **, $P < .01$; ****, $P < .0001$. Solid red line represents the median; dotted red lines represent the upper and lower quartiles. Data underlying this figure are available on figshare: https://figshare.com/projects/Bruckner_et_al_Data/136756.

Factor analysis reduced the measured dimensions into 3 factors that explain the most variance and plotting each neuron according to its score for the first and second factors revealed morphological similarity between neurons (Fig 3H and 3J). Two general categories of vTel$^{y321}$ neurons are apparent at 7 dpf: the vast majority that have simple arbors with few, short branches (Fig 3G and 3H: grey, blue) and a smaller subset that have long neurites with complex branching patterns (Fig 3G and 3H: orange, purple). Neurons in these broad classes are further subdivided into smaller morphological clusters. vTel$^{y321}$ neurons from GF larvae are overrepresented in clusters defined by increased morphological complexity, located on the right side of the dendrogram and factor analysis plots (Fig 3G and 3H). Complex arbors remain overrepresented in 14 dpf XGF larvae relative to their CVZ siblings (Fig 3I and 3J: orange, purple). Therefore, it appears that the majority of vTel$^{y321}$ neurons analyzed in GF or XGF fish are morphologically similar to those from CVZ siblings but that the microbiota normally restrains arborization of a subset of vTel$^{y321}$ neurons that become dramatically more complex in GF or XGF conditions.

To assess how the microbiota rearranges vTel$^{y321}$ neurites relative to the rest of the forebrain, we adapted existing tools and applied an automatic, signal-based pipeline to register individual vTel$^{y321}$ neurons to a reference vTel$^{y321}$ nucleus (S2 Fig and S3–S6 Movies). We expected that vTel$^{y321}$ subtypes might express some degree of spatial arrangement according to arbor morphology, perhaps with simple neuron somata clustered near the midline and complex neuron somata arranged at the periphery to project their neurites into adjacent functional regions. However, vTel$^{y321}$ somata do not appear organized in obvious patterns based on neurite morphology. As expected, the most complex vTel$^{y321}$ subtypes (orange) project their neurites predominantly into neuropil regions surrounding the vTel$^{y321}$ nucleus, and simple vTel$^{y321}$ arbors often remain within the vTel$^{y321}$ nucleus. At both 7 dpf (S2A Fig and S3 and S4 Movies) and 14 dpf (S2B Fig and S5 and S6 Movies), vTel$^{y321}$ neurites are significantly less dense in CVZ controls than in larvae raised GF or XGF, respectively. Viewed laterally (bottom panels in S2A and S2B Fig), it is apparent that the microbiota is required for vTel$^{y321}$ neurites to reach their normal targets in the ventral portion of the anterior commissure rather than the dorsal anterior commissure targeting observed in both GF and XGF larvae.

## Diverse bacterial strains promote forebrain microglial abundance and restrain vTel$^{y321}$ neurite density

We hypothesized that the microbiota might restrain vTel$^{y321}$ arborization via microglia, the brain's resident immune cells that regulate neurite outgrowth and pruning [55]. Though social phenotypes are not expressed until 14 dpf, our experiments provide evidence that the microbiota modulates vTel$^{y321}$ neuronal morphology as early as 7 dpf. If the microbiota exerts this influence by modulating development of forebrain microglial populations, then altered

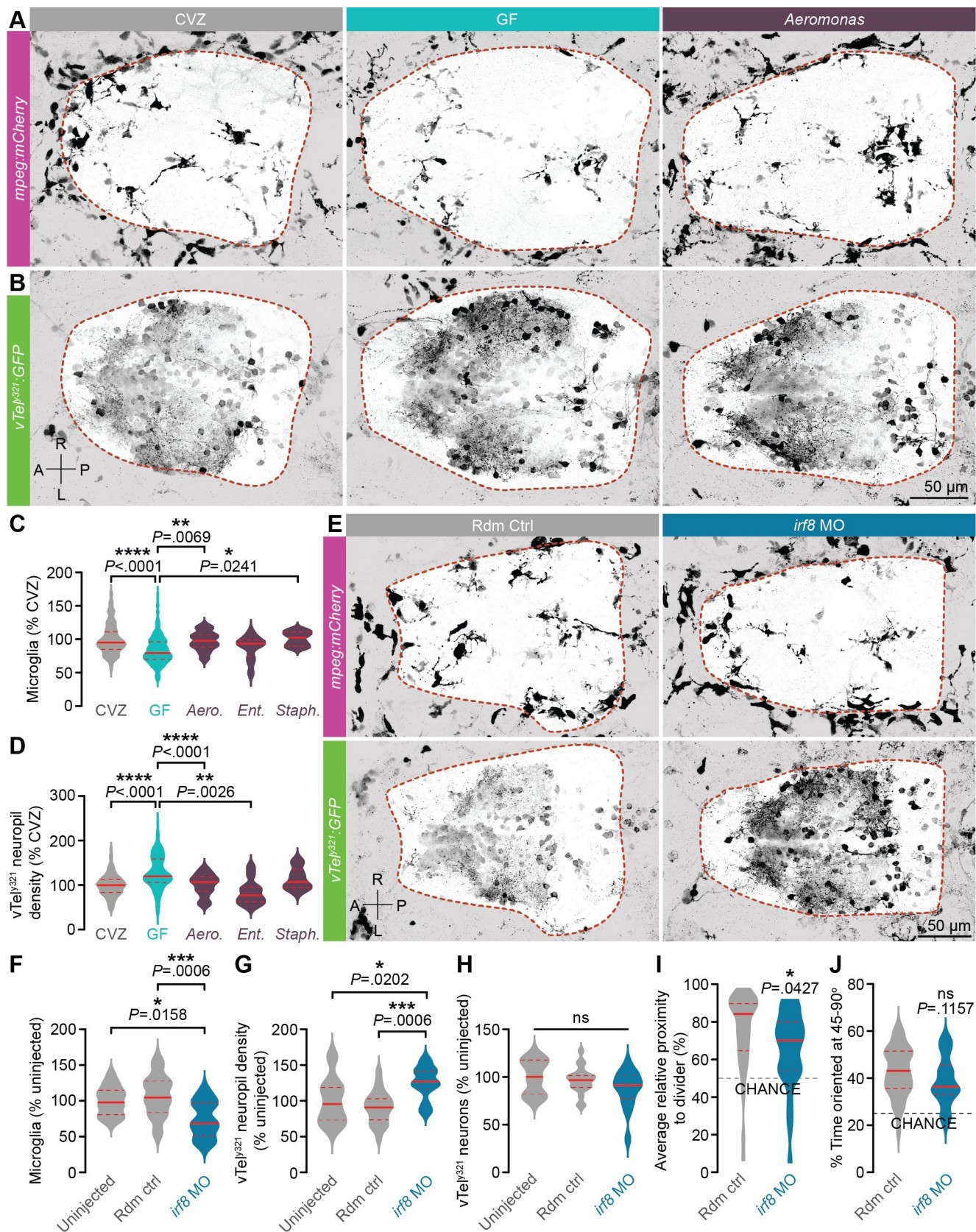

**Fig 4. Diverse bacterial strains promote microglial infiltration and restrain vTel$^{y321}$ neurite density.** (**A**, **B**) Representative dorsal views of maximum-intensity projections of *mpeg1:mCherryTg* (**A**; microglia and macrophages, magenta) and vTel$^{y321}$ GFP (**B**; neurons, green) from 7 dpf larvae that are CVZ (gray), GF (aqua), or mono-associated with *Aeromonas veronii* (purple; strain ZOR0001). Dotted lines indicate approximate forebrain boundary, segmented from the corresponding brightfield image. (**C**) The number of forebrain microglia, normalized to total forebrain volume, is decreased in GF larvae relative to CVZ larvae. This defect is rescued by mono-association with *Aeromonas veronii* (*Aero.*; strain ZOR0001) and *Staphylococcus* sp. (*Staph.*; strain ZWU0021) (*n* = 84 CVZ, 92 GF, 23 *Aero.*, 10 *Staph.*, and 11 *Ent.* larvae; Kruskal–Wallis test with Dunn's multiple comparisons). (**D**) vTel$^{y321}$ neuropil density is increased in GF larvae relative to CVZ larvae, and this defect is rescued by mono-association with *Aeromonas veronii* (*Aero.*; strain ZOR0001) and *Enterobacter cloacae* (*Ent.*; strain ZOR0014) (*n* = 105 CVZ, 98 GF, 32 *Aero.*, 10 *Staph.*, and 11 *Ent.* larvae; Brown–Forsythe and Welch ANOVA tests). (**E**) Dorsal views of maximum-intensity projections of vTel$^{y321}$ GFP (neurons, green) and *mpeg1:mCherryTg* (microglia and macrophages, magenta) in 7 dpf larvae injected at the 1-cell stage with either random control morpholino (Rdm Ctrl, gray) or *irf8* translation-blocking morpholino (*irf8* MO, blue). (**F**) Forebrain microglia, normalized to total forebrain volume, are fewer in *irf8* MO larvae relative to uninjected or random control–injected siblings (*n* = 17 uninjected, 22 random control, and 15 *irf8* MO larvae; one-way ANOVA with Tukey's multiple comparisons test). (**G**) vTel$^{y321}$ neuropil density is increased in *irf8* MO larvae relative to uninjected or random control–injected siblings (*n* = 17 uninjected, 22 random control, and 15 *irf8* MO larvae; one-way ANOVA with Tukey's multiple comparisons test). (**H**) The total number of vTel$^{y321}$ neurons is similar in *irf8* MO larvae relative to uninjected or random control–injected siblings (*n* = 17 uninjected, 22 random control, and 15 *irf8* MO larvae; one-way ANOVA with Tukey's multiple comparisons test). (**I**) Average relative proximity to the transparent divider is significantly reduced in *irf8* MO larvae relative to random control–injected siblings (*n* = 39 random control and 34 *irf8* MO larvae; unpaired *t* test). (**J**) Percent of time oriented at 45–90° is not significantly different in *irf8* MO larvae relative to random control–injected siblings (*n* = 39 random control and 34 *irf8* MO larvae; unpaired *t* test). ns, not significant; *, $P < .05$; **, $P < .01$; ***, $P < .001$; ****, $P < .0001$. Solid red line represents the median; dotted red lines represent the upper and lower quartiles. Data underlying this figure are available on figshare: https://figshare.com/projects/Bruckner_et_al_Data/136756.

microglia should be apparent in 7 dpf GF larvae. To test this hypothesis, we compared forebrain microglia of GF *mpeg1:mCherryTg* larvae to those of CVZ sibling controls (Fig 4A and 4C). The *mpeg:mCherry* transgene is expressed in both microglia and circulating macrophages, so we used brightfield images to semiautomatically segment the CNS boundary and distinguish these cell types (orange dotted line). Initial microglial accumulation in the zebrafish CNS largely occurs before microbiota colonization, so developmental delay between GF and CVZ larvae should not affect microglial establishment in the brain [28,56]. However, since microglia distribution remains dynamic after colonization and some GF larvae are smaller than their CVZ siblings, microglial counts were normalized to forebrain volume. Forebrain microglia are significantly fewer in GF larvae relative to CVZ controls, reinforcing the conclusion that the microbiota is required for normal microglial abundance in the zebrafish forebrain (Fig 4A and 4C).

We hypothesized that specific bacterial strains restrict vTel$^{y321}$ arborization and promote microglial abundance. To assess this, we developed a pipeline for high-throughput screening of previously isolated zebrafish-associated bacterial strains [47]. In fish raised CVZ (gray), GF (aqua), or derived GF and mono-associated at day 0 with individual bacterial strains (purple), we simultaneously imaged mCherry-expressing microglia and GFP-expressing vTel$^{y321}$ neurons and, using intensity-based thresholding of vTel$^{y321}$ neurons and a brightfield image to segment the forebrain surface, quantified neuropil density in 3D (Fig 4A–4D). As expected, total vTel$^{y321}$ neuropil density is significantly increased in GF larvae relative to CVZ siblings at 7 dpf (Fig 4B and 4D). Interestingly, mono-association with gram-negative *Aeromonas veronii* (strain ZOR0001) or *Enterobacter cloacae* (strain ZOR0014) and with gram-positive *Staphylococcus* sp. (strain ZWU0021) all at least partially restore forebrain microglial abundance and vTel$^{y321}$ neuropil density defects in GF larvae, though the effects of *Enterobacter* on microglial abundance and the effects of *Staphylococcus* on vTel$^{y321}$ neuropil density are trending but not statistically significant. These results suggest that a general feature common among diverse microbial taxa influences zebrafish forebrain neurodevelopment.

## Microglia are required for vTel$^{y321}$ neuronal arborization

Microglia adopt diverse roles throughout neurodevelopment, starting as regulators of neuronal cell death, axon outgrowth, and fasciculation during early development and transitioning to steady-state regulation of synapse maturation, function, and pruning [19]. Our results suggest

that the microbiota may restrain vTel$^{y321}$ arbor complexity and targeting by promoting the abundance of forebrain microglia available to remodel vTel$^{y321}$ arbors. Indeed, microglial depletion in the murine hippocampus increases spine density of hippocampal CA1 neurons and impairs social behavior [21]. However, it is also possible that zebrafish forebrain microglia are not remodeling vTel$^{y321}$ arbors at this time point and that the microbiota therefore influences microglial abundance and vTel$^{y321}$ arbor complexity independently. To address this possibility, we reduced forebrain microglia by injecting embryos with a previously validated morpholino against the microglial gene *irf8* [57] and measured microglial abundance and vTel$^{y321}$ neuropil density at 7 dpf as described above (Fig 4E). As observed previously [57], *irf8* morpholino significantly reduces microglia relative to uninjected or random control–injected animals (Fig 4F). Notably, the reduction in microglia that we observe in 7 dpf *irf8* morphants is not as dramatic as reported at 3 dpf, potentially due to waning morpholino efficacy or local microglial proliferation [57,58]. Yet, vTel$^{y321}$ neuropil density was also increased in *irf8* morphants relative to uninjected or random control–injected siblings, providing strong evidence that microglia restrain vTel$^{y321}$ arborization during early larval development (Fig 4G). *irf8* morphants have essentially the same number of vTel$^{y321}$ neurons as uninjected or random control–injected animals (Fig 4H), suggesting that microglia are not significantly involved in forebrain neuronal apoptosis at 7 dpf. To our knowledge, this is the first demonstration that microglia are required for normal neuronal arborization in zebrafish larvae. To address whether microglial remodeling of vTel$^{y321}$ arborization is required for normal social behavior, we assayed social behavior phenotypes as described above in 14 dpf larvae in which microglia have been reduced with morpholino against *irf8*. Compared to random control–injected siblings, *irf8* morphants spent significantly less time in close proximity to the stimulus fish (Fig 4I). Though *irf8* morphants also appeared to spend slightly less time than random control–injected siblings oriented at 45 to 90° relative to the stimulus fish, this difference was not statistically significant (Fig 4J). As for XGF fish, impaired social behavior in *irf8* morphants cannot be explained by impaired larval motility (% time in motion; random control, $39.92 \pm 2.07$, $n = 39$ larvae; *irf8* MO, $36.32 \pm 2.22$, $n = 34$ larvae; $P = .2391$, unpaired *t* test).

Further exploring the link between forebrain microglia and neuronal arborization, we used the approach described above to compare the depth within the forebrain of microglia and vTel$^{y321}$ arbors in CVZ and GF larvae (S3A Fig). The defects in microglial abundance and neuropil density described above are recapitulated (S3B and S3C Fig), and the dorsal shift in vTel$^{y321}$ neuropil density that we observed in individually segmented neurons from GF larvae is also apparent when the entire population is quantified, as the 3D center of mass of the vTel$^{y321}$ neuropil is shifted dorsally in GF larvae relative to CV controls (S3D Fig). The average dorsoventral position of forebrain microglia in GF larvae is also significantly reduced relative to microglia in CVZ larvae (S3E and S3F Fig). Together, these findings suggest that the microbiota may restrain dorsal vTel$^{y321}$ neurite targeting by specifically influencing the number of microglia in dorsal forebrain territories.

## The microbiota does not influence forebrain microglial morphology or dynamics

It is possible that in addition to promoting microglial localization, the microbiota also promotes microglial phagocytic activity, which is required for their role in responding to local insult, clearing apoptotic material, and for normal synaptic pruning and maturation during brain development [19]. Microglia are traditionally classified as either "ramified" or "amoeboid." Ramified microglia do not travel through the tissue but scan relatively stable territories with dynamic processes that monitor, maintain, and prune synapses [59]. Amoeboid microglia

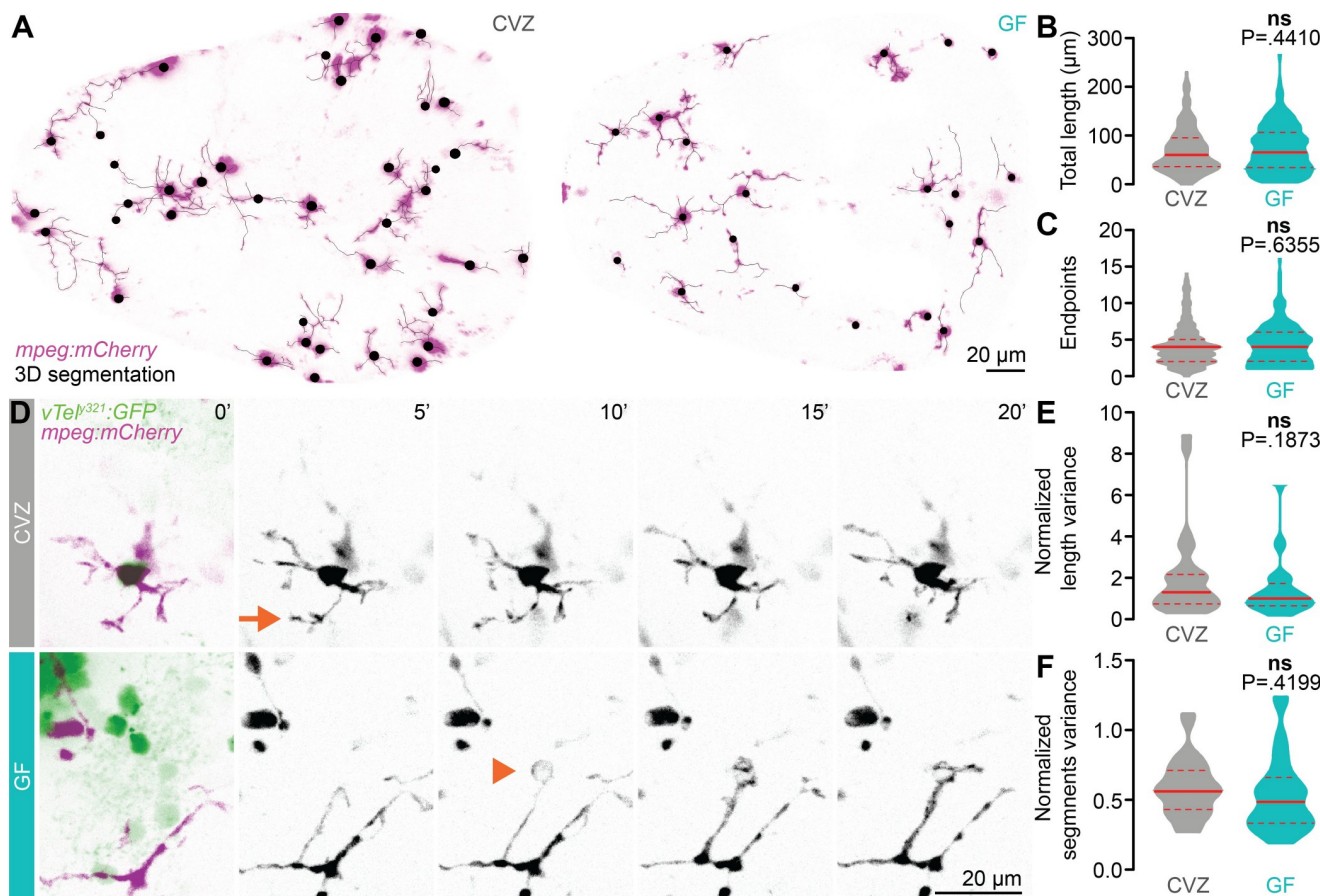

**Fig 5. The microbiota does not influence forebrain microglial morphology or dynamics.** (**A**) Maximum-intensity Z-projections of representative *mpeg1*: *mCherryTg*-positive microglia (magenta), segmented in 3D for morphological quantification (black), from CVZ (left) and GF (right) larvae. The *mpeg1*: *mCherryTg* channel is masked using a brightfield image to remove macrophages from the analysis. (**B**, **C**) Total microglial length (**B**) and number of endpoints (**C**) are similar in GF larvae and CVZ siblings (*n* = 295 microglia from 8 CVZ larvae 204 microglia from 8 GF larvae; Mann–Whitney *U* tests). (**D**) Maximum-intensity Z-projections of representative *mpeg1:mCherryTg*-positive microglia (magenta) and vTel$^{y321}$ neurons (green), from CVZ (gray) and GF (aqua) larvae, every 5 minutes across a 20-minute time series. Arrow indicates a rapidly retracting protrusion, and arrowhead indicates an extending protrusion that likely envelops an unlabeled neuronal soma. (**E**, **F**) Mean-normalized length variance (**E**) and mean-normalized segments variance (**F**) of vTel$^{y321}$-embedded microglia across the time series are similar in GF larvae and CVZ siblings (*n* = 28 microglia from 4 CVZ larvae and 28 microglia from 3 GF larvae; unpaired *t* test for mean-normalized segments variance; Mann–Whitney *U* test for mean-normalized length variance). Solid lines represent the mean; dotted lines represent SEM. ns, not significant. Solid red line represents the median; dotted red lines represent the upper and lower quartiles. Data underlying this figure are available on figshare: https://figshare.com/projects/Bruckner_et_al_Data/136756.

retract many of their processes, can proliferate, and migrate through tissue in response to infection or injury. As microglia executing these activities have distinct morphologies, measurements of microglial morphology can be used to assess the proportion of the microglial population available for each function. We used semiautomated fluorescence-based segmentation to quantify microglial morphology (Fig 5A). In 7 dpf GF larvae and CVZ controls, we observed diverse morphologies that include ramified microglia with long, complex branching patterns and amoeboid microglia with larger cell bodies and fewer branches (Fig 5A). However, we did not observe significant differences in microglial total length or number of endpoints between GF and CVZ larvae (Fig 5B and 5C). Though this suggests that the microbiota does not influence forebrain microglial morphology, microglial process dynamics have also been linked to microglial activity surveilling the surrounding tissue and remodeling synapses [59–61]. To test the hypothesis that the microbiota restrains vTel$^{y321}$ arbor density by

influencing microglial process dynamics without affecting the morphological subtypes present, we segmented *mpeg1:mCherryTg*-positive microglia imaged live during 20-minute spinning disk confocal volumetric time series (Fig 5D and S7 and S8 Movies). Across the time series, we did not observe significant differences in microglial kinetics in GF and CVZ larvae (Fig 5E and 5F and S7 and S8 Movies).

## The microbiota influences microglial gene expression

As brain-resident immune cells that shape neurodevelopment, microglia are uniquely positioned to receive molecular input from the microbiota and modulate neural circuits. Though the microbiota is required for forebrain microglial abundance and vTel$^{y321}$ arborization, it remains unclear how these phenotypes are linked at the molecular level. To identify candidate microglial genes that are modulated by the microbiota and regulate microglial function during early brain development, we identified microglia in an existing single-cell RNA sequencing (scRNAseq) dataset from 6 dpf larvae raised either GF or CVZ [62]. Reclustering the 392 cells in Cluster 36 [62], which is the only *mpeg1.1$^+$* immune cell cluster and therefore likely includes macrophages and microglia, generated 9 new subclusters (Fig 6A). Cells from CVZ and GF larvae are evenly distributed throughout the 9 clusters (S4A Fig), consistent with our morphological analysis and indicating that the microbiota does not affect the relative abundance of each cell type.

Next, we sought to determine whether this clustering accurately separates microglia and macrophages. The common developmental origin and extreme transcriptional similarity between microglia and macrophages decreases the utility of individual "marker" genes, even when used in multi-gene combinations [19]. To address this issue, we mapped expression of a 75-gene core microglial fingerprint to our 9 clusters (Fig 6B and S1 Table). Critically, though individual genes in this list may be expressed in both macrophages and microglia, combinatorial expression of the entire fingerprint is unique to microglia. A significantly greater percentage of cells (for many genes >80%) in subclusters 1, 2, and 4 express the microglial fingerprint at higher levels than the other clusters, indicating that the cells in these clusters are predominantly microglia. As they also express *mpeg1.1*, cells in the remaining clusters are likely macrophages. However, our searches for known markers of macrophage subsets, including border-associated, M1, and M2 macrophages, did not reveal unique identities among the macrophage clusters [63–65]. It is possible that these macrophage clusters represent biologically relevant cell types or states, but the lack of pro-inflammatory challenge in this dataset likely minimizes the representation of activated macrophages and small sets of qualitative markers may be insufficient to distinguish macrophage subtypes.

Focusing on microglial clusters 1, 2, and 4, we hypothesized that they represent ramified and amoeboid microglial subtypes and examined expression of previously described markers for ramified and amoeboid microglia in the zebrafish brain [66]. A large percentage of Cluster 1 cells express amoeboid microglial markers at high levels, especially *ccl34b.1*, *lgals9l3*, *lgals3bpb*, *apoc1*, and *apoeb* (Fig 6C). Cluster 4 cells, on the other hand, do not strongly express these amoeboid microglial markers and instead express a ramified microglia signature including *aif1l*, *cmklr1*, and *ccl35.2* (Fig 6C). Cluster 1 and Cluster 4 are tightly linked in UMAP space with several cells of each cluster overlapping, likely corresponding to the ability of larval zebrafish microglia to rapidly cycle between ramified and amoeboid states (Fig 6A) [58]. Hierarchical clustering suggests that Cluster 1 amoeboid microglia are more closely related to Cluster 2 cells than the ramified microglia in Cluster 4 (Fig 6A). Though they do not express an amoeboid or ramified molecular signature, Cluster 2 microglia strongly express proliferative markers (*mki67*, *pcna*) that are not expressed in the amoeboid or ramified microglia of

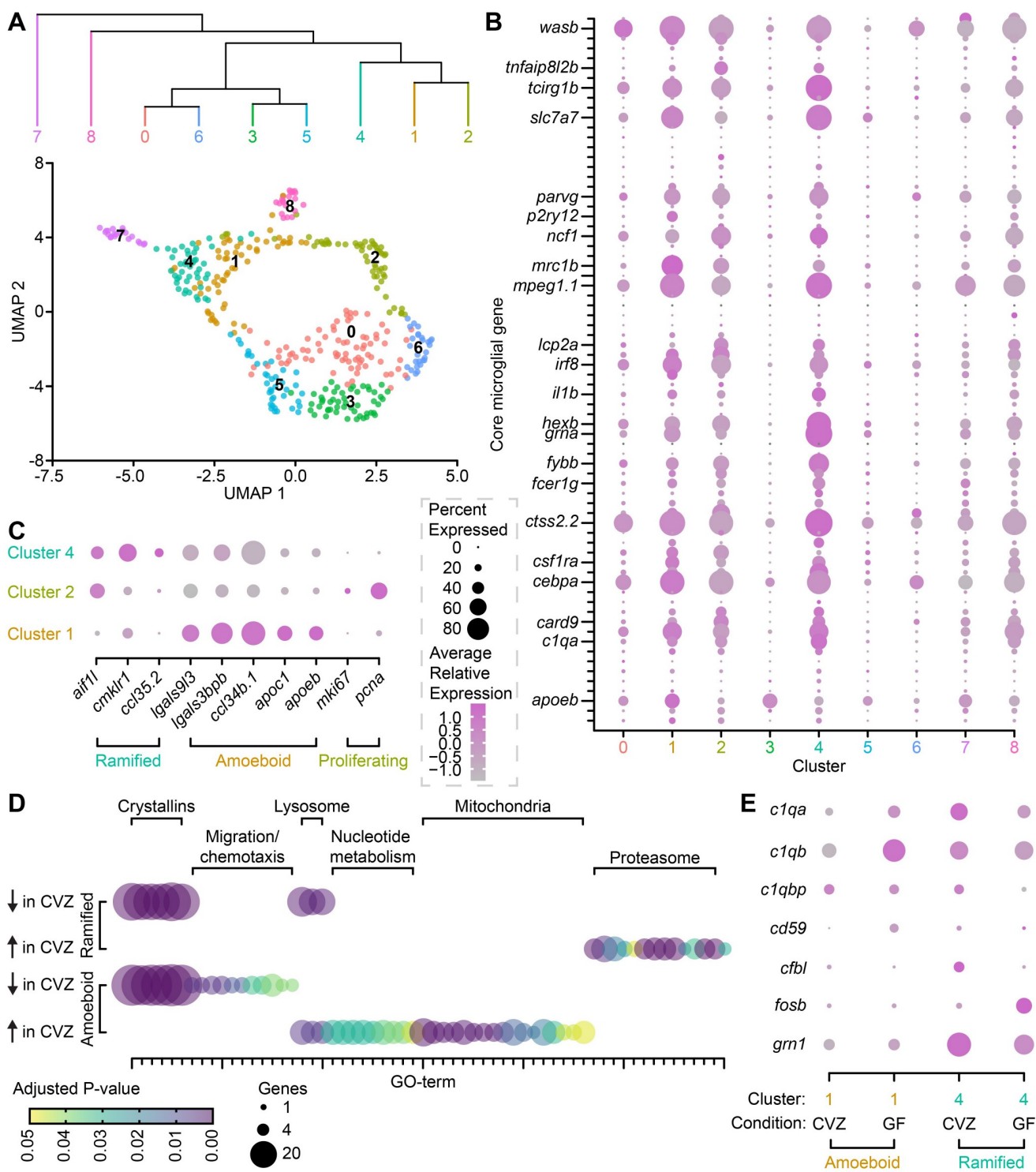

**Fig 6. The microbiota influences microglial gene expression.** (**A**) UMAP visualization and hierarchical clustering of immune cells from *mpeg1.1*[+] Cluster 36 in Massaquoi and colleagues [62]. (**B**) Average relative expression level (color) and percent of clustered cells (dot size) expressing each member of a 75-gene microglial fingerprint across 9 clusters of *mpeg1.1*[+] Cluster 36 immune cells. Select transcripts are labeled and the complete fingerprint is included in S1 Table. (**C**) Average relative expression level (color) and percent of clustered cells (dot size) expressing ramified, amoeboid, and proliferative microglial markers in clusters 1, 2, and 4. (**D**) Number of included genes (dot size) and adjusted *P* value (color) for gene ontology (GO) terms expressed in ramified microglia (cluster 4, top) and amoeboid microglia (cluster 1, bottom). GO term identities are included in S2 Table. (**E**). Average relative expression level (color) and percent of amoeboid (cluster 1, orange) or ramified (cluster 4, teal) cells (dot size) from CVZ or GF larvae that express complement pathway transcripts. Data underlying this figure are available on figshare: https://figshare.com/projects/Bruckner_et_al_Data/136756 and in S1–S5 Tables.

Clusters 1 and 4 (Fig 6C). Larval zebrafish microglia proliferate by adopting an amoeboid morphology, dividing, and rapidly extending ramified processes again [58], so it is unsurprising to observe a large proliferative population of microglia closely linked to amoeboid microglia in UMAP space.

Having identified microglial clusters in our scRNAseq data, we used a gene ontology (GO) approach to assess whether the microbiota influences gene expression in these cells (Fig 6D and S2 and S3 Tables). In both amoeboid and ramified microglia, the microbiota restrains expression of genes in the expansive *crystallin* family. As described in Massaquoi and colleagues [62], increased expression of highly stable Crystallin proteins across most cell types in GF larvae suggests globally more quiescent cell states. In amoeboid microglia, the microbiota restrains expression of migration and chemotaxis genes while promoting expression of genes linked to lysosomal function, nucleotide metabolism, and mitochondrial function (Fig 6D, bottom). In ramified microglia, the microbiota restrains expression of lysosomal genes while promoting proteasome gene expression. In these cells, the microbiota also promotes expression of a handful of genes in unique GO terms including "defense response to Gram-positive bacterium," "regulation of I-kB kinase/NF-kB signaling," and "peptidoglycan muralytic activity," suggesting that microglial responses to microbial signaling may also be affected.

## The microbiota promotes *c1q* expression

Our GO analysis suggests that ramified and amoeboid microglia have altered lysosomal and proteasomal function. We also detected microbial modulation of other microglial pathways that facilitate arbor remodeling, especially the complement pathway [67] (S4 and S5 Tables). Zebrafish express orthologues of all mammalian complement components [68], and we observed differential expression of *c1qa*, *c1qb*, *c1qbp*, *cd59*, *cfbl*, *fosb*, and *grn1* in GF relative to CVZ microglia (Fig 6E). Complement pathway genes are more strongly expressed in ramified microglia than amoeboid microglia (Fig 6E), which is unsurprising as ramified microglia are thought to be the primary synaptic and axonal sculptors [64]. Except for *fosb*, expression of each gene decreases in GF relative to CVZ ramified microglia, suggesting that the normal role of the microbiota is to promote complement expression in these cells (Fig 6E, teal). In amoeboid microglia, *c1qbp*, *cfbl*, *fosb*, and *grn1* expression is not significantly affected by the microbiota, but expression of *cd59*, *c1qa*, and *c1qb* increases in GF relative to CVZ amoeboid microglia (Fig 6E, orange). Therefore, though the microbiota clearly promotes expression of a molecular pathway that regulates neurite remodeling in ramified microglia, there may also be complex compensatory effects in other cell types and states.

As our scRNAseq data represent a temporal snapshot of expression across the organism, we used fluorescence in situ hybridization RNA labeling to test whether *c1qa* expression is affected by the microbiota specifically in 7 dpf larval forebrain microglia (Fig 7A). We did not detect any labeling in either group with a probe against *c1qb* but cannot exclude inefficient probe binding. Though, as we observed previously, microglial size is unchanged (Fig 7B), the average *c1qa* intensity in each microglial cell and average *c1qa* punctum intensity are significantly reduced in GF larvae relative to CVZ siblings (Fig 7C and 7D). Many of the microglia we imaged did not have any *c1qa* puncta (Fig 7A–7C); however, the number of *c1qa* puncta is positively correlated with microglial volume. This may represent the phenotypic continuum from amoeboid to ramified microglia, which is weaker in microglia from GF larvae (CVZ, $R^2$ = .1271; GF, $R^2$ = .0465). These data reinforce the changes in complement gene expression we detected with scRNAseq and, with the experiments described above, suggest that microbial modulation of microglial abundance and function converge to affect social behavior by restraining forebrain neuronal arbor remodeling.

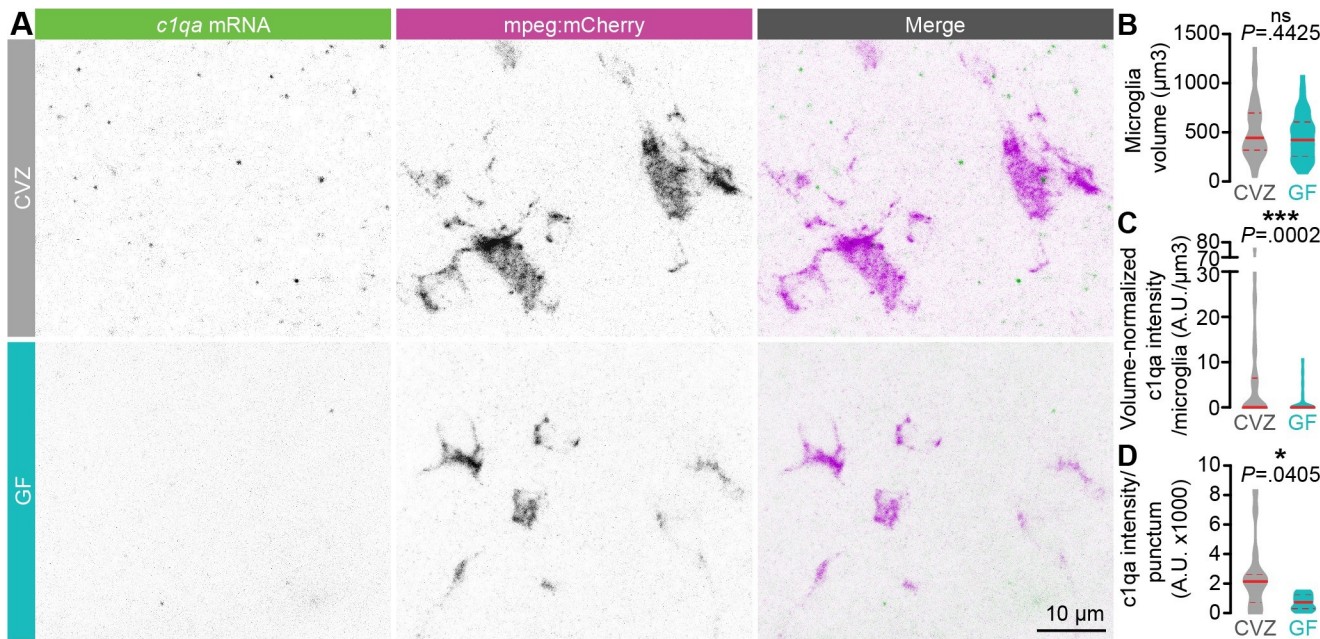

**Fig 7. c1q expression is promoted by the microbiota.** (**A**) Representative maximum intensity projections of fluorescence in situ hybridization against *c1qa* (green) and antibody labeling against *mpeg1:mCherryTg* (microglia, magenta) in the forebrain of 7 dpf larvae raised GF or CVZ. (**B**) Microglial volume is similar in GF and CVZ larvae. (**C**) *c1qa* intensity per microglia, normalized by microglial volume, is decreased in microglia in GF larvae relative to microglia in CVZ larvae (*n* = 36 microglia from 14 CVZ larvae and 51 microglia from 17 GF larvae; Mann–Whitney *U* tests). Note that *c1qa* signal was undetectable in some microglia, more often in larvae raised GF than CVZ. (**D**) *c1qa* intensity per punctum is decreased in microglia in GF larvae relative to microglia in CVZ larvae (*n* = 18 *c1qa* puncta from 14 CVZ larvae and 7 *c1qa* puncta from 17 GF larvae; Mann–Whitney *U* test). ns, not significant; *, *P* < .05; ***, *P* < .001. Solid red line represents the median; dotted red lines represent the upper and lower quartiles. Data underlying this figure are available on figshare: https://figshare.com/projects/Bruckner_et_al_Data/136756.

## Discussion

Our key finding is that the microbiota is required during an early period of development for zebrafish to establish normal social behavior that only manifests at a later developmental stage. This conclusion is the result of several different lines of evidence. Social behavior requires the vTel[y321] nucleus, which is homologous to "social nuclei" in other vertebrate models. We found that the microbiota significantly alters projections of a subset of vTel[y321] neurons by restraining complexity of their neurites. We also found that the microbiota is required for infiltration of the appropriate number of microglia into the vTel[y321] brain region and that these microglia refine vTel[y321] neurites during early development. We found that a diverse set of individual zebrafish-associated bacterial strains are sufficient for normal forebrain microglial abundance and vTel[y321] neurite density. We also found that the microbiota influences microglial gene expression, including tuning levels of neurite remodeling genes in the complement pathway. We discuss each of these discoveries in turn.

We observed that the microbiota is required early for normal social behavior exhibited at least a week later, suggesting that the microbiota influences social behavior by modulating neurodevelopment during an early sensitive period. Zebrafish social phenotypes increase in complexity significantly from 10 to 16 dpf [5]. Our observations suggest that the neuronal circuitry that facilitates this behavior requires microbial input as it is being built, likely between when the gut becomes patent at 4 dpf and when we assayed neuroimmune phenotypes and colonized with a normal microbiota at 7 dpf. A previous study identified an *L. rhamnosus* strain that can promote shoaling in otherwise conventionally raised zebrafish [43], but, to our knowledge,

ours is the first study to demonstrate that an intact microbiota is required for normal zebrafish social behavior such as conspecific orienting (Fig 1). These findings are consistent with the association between altered gut microbiota composition and neurodevelopmental disorders including ASD [2]. The zebrafish microbiota is also required for normal activity and anxiety-like behavior, suggesting that these phenotypes could be linked to the impaired social behavior we observe [45,69]. However, GF larvae and their CVZ siblings perform similarly in our opto-motor assay and motor behavior remains unaffected in 14 dpf XGF larvae; thus, the social defects we observe in XGF larvae may instead result from impaired integrative circuits downstream of sensory input and upstream of motor output. Altered anxiety-like behavior has also been reported in GF mice, so future experiments could explore whether microbial modulation of anxiety and social behavior intersect in brain regions such as the subpallium, which in zebrafish includes vTel$^{y321}$ neurons [70]. Previous work in other organisms has largely focused on how the microbiota influences adult neurodevelopmental phenotypes; therefore, we decided to take advantage of the strengths of zebrafish as a model to understand how the microbiota influences social behavior specifically during early forebrain neurodevelopment.

As the microbiota is required early for later-developing social behavior, we postulated that it influences early neurodevelopmental events specifically in circuits that regulate social behavior. Therefore, we focused on whether the microbiota is required for normal development of subpallial vTel$^{y321}$ neurons, which are required for social behavior [6]. This region of the zebrafish brain is thought to be an integrative part of a circuit homologous to subpallial regions of the mammalian brain that also regulate social behavior, including the lateral septum, preoptic area, and hypothalamus [13–15]. Whether development of these regions, which regulate murine social behavior, is also influenced by microbial signals has not been investigated. However, several previous studies found that the microbiota is required for normal dendrite morphology in the murine anterior cingulate cortex, amygdala, and hippocampus [71,72], so it seems likely that microbial modulation of social behavior in other vertebrates could also occur by modulation of circuit connectivity as we observed in zebrafish vTel$^{y321}$ neurons. How the microbiota influences neurodevelopment outside of the subpallium bears investigation in future studies. Based on the diverse neuronal and microglial phenotypes that have been reported in GF rodents [4,49,71–75], it is reasonable to expect that the microbiota modulates a variety of neurodevelopmental mechanisms in brain regions other than the subpallium.

The changes in neuronal morphology we observe could be, at least in part, a downstream effect of microbial modulation of gene expression in vTel$^{y321}$ neurons, as the microbiota has widespread impact on gene expression in the zebrafish CNS [62]. Though studies examining the microbial modulation of neuronal gene expression in rodents have largely focused on the cortex and hippocampus, altered amygdala expression of genes including *BDNF* and multiple neurotransmitter pathways could also affect social behavior by modifying neuronal morphology or connectivity [35,41,70,76–78]. For example, oxytocin signaling in the murine hypothalamus is altered following probiotic treatment [79–81]. As hypothalamic oxytocin is also important for zebrafish social behavior, it will be interesting to investigate whether microbial modulation of social behavior, neuronal gene expression, and cytoarchitecture intersect in forebrain neuromodulatory systems [82,83].

Neuronal morphology is the foundation for circuit connectivity and function, so our finding that the microbiota influences the morphology and targeting of subpallial vTel$^{y321}$ neurons provides strong evidence that the microbiota normally plays a critical role in establishing social circuitry. The exuberant arborization we observe early in GF fish persists even with an additional week of development in the presence of an intact microbiota, suggesting that vTel$^{y321}$ connectivity impaired during early development results in persistently miswired circuits. Microbial modulation of neurite complexity appears critical for normal ventral targeting of

many vTel[y321] neurites; we observed GF or XGF vTel[y321] neurites extending toward dorsal destinations rather than extending ventrally as in CVZ controls. It will be interesting to further investigate the functional consequences of vTel[y321] neurite ventral targeting and the mechanisms by which the microbiota normally modulate that targeting. Dopaminergic neurons and projections populate the most ventral aspects of the zebrafish subpallium and synapse with vTel[y321] neurites that project ventroposteriorly toward the anterior commissure, raising the exciting possibility that the microbiota influences social behavior by promoting connectivity between vTel[y321] neurons and monoamine circuits known to regulate social reward [84–87].

Since microglia are ideally positioned to both receive microbial signals and modify neurons, we hypothesized that the microbiota promotes social behavior by influencing development or function of microglia that modify vTel[y321] neurites. There is precedence for this idea in the literature; murine microbiota disruption impairs responses to social novelty and alters microglial morphology and reactivity in the hippocampus and cortex [73,77]. Microglia in GF mice are also metabolically dysfunctional, larger, less mature, less responsive to LPS challenge, and more abundant in the cortex, corpus callosum, hippocampus, olfactory bulb, and cerebellum than in specific pathogen-free (SPF) controls [49,74,75]. Contrary to these murine phenotypes, zebrafish forebrain microglia are reduced in GF larvae relative to CVZ controls and forebrain microglial morphology does not appear influenced by the microbiota. We also do not observe an effect on the ratio of amoeboid and ramified microglia. How can these results be reconciled? One possibility is that the microbiota has differential effects on microglia across the brain, as microglial subtypes are heterogeneously distributed [64]. Microglia are drawn into the brain by chemokine signaling and neuronal apoptosis before 4 dpf, prior to microbiota colonization [28,56,88]. The number of vTel[y321] neurons is essentially the same in GF and CVZ fish, microglial reduction does not affect the number of vTel[y321] neurons, and apoptotic neurons are largely absent by 6 dpf [56], so it is unlikely that the microbiota draws microglia to the developing forebrain by promoting apoptosis. It is also possible that the microbiota has differential effects on early larval and juvenile microglia, which have distinct developmental origins and neuroimmune functions [25]. Further study will be necessary to gain a deeper understanding of the role of the microbiota in promoting or suppressing microglial development or dynamics and how this varies across brain regions, developmental time, and taxa.

The three microglial clusters we identify likely include amoeboid, ramified, and proliferative microglia and represent previously identified microglial heterogeneity across the zebrafish brain [64,66]. Our *ccl34b.1*[+] population ("amoeboid" cluster 1) likely corresponds to optic tectum-enriched neurogenic associated microglia (NAMs) and our *ccl34b.1*[−] population ("ramified" cluster 4) likely corresponds to hindbrain-enriched synaptic region–associated microglia (SAMs) [64]. Separating microbial modulation of gene expression in distinct microglial functional subtypes enables us to identify homeostatic, neuroinflammatory, and parainflammatory effects of the microbiota [89]. In amoeboid cluster 1 NAMs, the microbiota restrains expression of migration and chemotaxis genes while promoting expression of genes linked to lysosomal function, nucleotide metabolism, and mitochondrial function. Combined with the change in Crystallin gene expression and impaired mitochondrial function that we and others describe [62,74], this supports the idea that the microbiota is critical for normal metabolic activity in amoeboid microglia. Our observation that the microbiota broadly promotes expression, in amoeboid microglia, of genes involved in mitochondrial oxidative phosphorylation is consistent with the idea that the microbiota supports microglial function in part by minimizing mitochondrial oxidative stress [49,62].

The microbiota has the opposite effect on lysosomal function in ramified microglia, where it also promotes expression of many proteasomal pathway genes. In addition to regulating cell homeostasis, the microglial proteasome is also involved in microglial activation in response to

injury or infection. This suggests that the microbiota partially suppresses ramified microglia reactivity, which is linked to multiple neurological disorders, perhaps by providing innate immune training [90,91]. Hypothesizing that the microbiota might influence vTel[y321] neurite remodeling via microglia resident in the forebrain neuropil, we identified a suite of complement pathway genes with microbiota-promoted expression in ramified cluster 4 SAMs. The complement system is a well-studied mechanism through which microglia remodel neurites and zebrafish express all complement components [68,92]. Contrary to the complement down-regulation we observe in ramified microglia from GF larvae, expression of multiple complement factors is increased in microglia isolated from adult GF mice relative to SPF controls [74,75]. A subsequent study observed decreased expression of complement components C1qbp and integrin subunit alpha X (Itgax) in microglia from newborn GF mice [93], so these results may represent a difference in microglial function between steady-state maintenance of the adult brain and early postnatal development. However, we also observe an up-regulation of complement pathway genes specific to amoeboid microglia that suggests the microbiota may not affect all functional microglial subtypes equally. Normally, amoeboid microglia are more abundant in neurogenic regions where, rather than remodeling synapses, they engulf apoptotic cell corpses [64]. It is intriguing to speculate that altered microbial signaling could blur the lines between functionally discrete microglial subtypes, resulting in a brain that is both profoundly miswired and unable to respond appropriately to future insult.

Mono-association of GF zebrafish with multiple commensal bacterial strains, including the gram-negative *Aeromonas veronii* strain ZOR0001 and *Enterobacter cloacae* strain ZOR0014, and gram-positive *Staphylococcus sp.* Strain ZWU0021, at least, partially restores defects in forebrain microglial abundance and vTel[y321] neurite density. This suggests that common microbial features shared between gram-negative and gram-positive bacterial strains can activate a host pathway evolved to respond to diverse microbes, such as the complement pathway. For example, microglial function and homeostasis phenotypes in adult GF mice can be restored by the common bacterial fermentation product acetate [74], though the dominance of facultative aerobic strains in the larval zebrafish microbiota suggests that the microbial modulation of microglia that we observe occurs by a different mechanism [47]. Not all mono-associated bacterial strains restored forebrain neurodevelopmental features to the same degree. This is consistent with the idea that distinct bacterial strains can elicit different degrees of innate immune activation, including complement signaling [94,95]. It is thus easy to imagine how distinct microbiota compositions could result in variable levels of microglial complement signaling and, therefore, synaptic pruning, predisposing some individuals to neurodevelopmental disorders such as ASD.

Effective intervention in diverse neurodevelopmental disorders requires understanding both intrinsic and extrinsic pathways that guide development. The neurodevelopmental processes that build social behavior across taxa are poorly understood. Our study reveals microbial modulation of social behavior in a model vertebrate well suited to simultaneous study of the microbiota, brain, and immune system [96] and provides the first in-depth look at how interactions among these components modulate circuit formation and maintenance and behavior.

## Materials and methods

### Ethics statement

All zebrafish experiments were approved by the University of Oregon Institutional Animal Care and Use Committee (protocols 18–08,18–29, and 20–15).

## Zebrafish lines and husbandry

All zebrafish lines were maintained as previously described at 28˚C with a 14/10 light/dark cycle [97]. AB × TU strain wild-type fish were raised CVZ and GF for behavior experiments. For vTel[y321] sparse mosaic labelling, *Tg(14xUAS-E1b:UBCi-blo-nls-emGFP-βglobin-blo-lyn-TagRFPT-afp)y562* (*UAS:bloSwitch*) and *Tg(myl7:GFP-hsp70l:B3r-2a-Cer)y560* (*hsp70l:B3*) lines gifted by the Burgess laboratory were crossed to *Et(REX2-SCP1:GAL4FF)y321* (*y321Et*) by maintaining a stable line heterozygous for *UAS:bloSwitch* and *y321Et*, which was then crossed to *hsp70l:B3* [53]. For simultaneous imaging of microglia and vTel[y321] neurons, homozygous *y321Et; UAS:GFP* fish were crossed to homozygous *Tg(mpeg1:mCherry)gl23* (*mpeg1:mCherryTg*). AB × TU, *y321Et*, *UAS:GFP*, and *mpeg1:mCherryTg* lines are available from the Zebrafish International Resource Center (ZIRC; http://zebrafish.org).

## Gnotobiology

Zebrafish embryos were raised GF, XGF, or CVZ as previously described [50,98]. Briefly, embryos were treated from 0 to 6 hours post fertilization (hpf) in embryo medium (EM) containing 100 μg/mL ampicillin, 250 ng/mL amphotericin B, 10 μg/mL gentamycin, 1 μg/mL tetracycline, and 1 μg/mL chloramphenicol. In a class II A2 biological safety cabinet, embryos were briefly surface-sterilized with 0.1% PVP-I and 0.003% sodium hypochlorite, washed with sterile EM, and transferred to 50 mL tissue culture flasks at a density of 1 fish/1 mL sterile EM. CVZ flasks were inoculated with 200 μl water from the parental tank immediately following the GF derivation procedure. Mono-associated larvae were generated as previously described [50], except that washed bacterial culture was added to GF flasks at day 0 at approximately $10^6$ CFU/mL. Inoculated strains are previously described and included *Aeromonas veronii* strain ZOR0001, *Enterobacter cloacae* strain ZOR0014, and *Staphylococcus* sp. strain ZWU0021 [47]. For flasks containing 7 dpf larvae, sterility was assessed by direct visualization of microbial contaminants with phase optics on an inverted microscope at 40× magnification once per day and by culturing media on LB agar at 28˚C for 2 days following terminal sampling. XGF larvae and CVZ siblings were inoculated with system water at 7 dpf and fed rotifers 3 times daily until terminal sampling at 14 dpf.

## Behavior

Social behavior was assessed with our previously published dyad assay for postflexion larval and adult zebrafish [5,6]. Briefly, AB × TU 14 dpf sibling pairs for each condition were placed in isolated custom-built acrylic tanks (50 mm width × 50 mm length × 20 mm depth) and allowed to interact for 10 minutes via adjoining transparent tank walls. Larvae were imaged from below at 10 fps using a Mightex SME-B050-U camera. The arena was illuminated from above with a white LED panel (Environmental Lights) with light-diffusing plastic as a tank lid to improve image quality. Fish that spent <10% of the experiment in motion (moving at least one-third of their total body length per frame) were not included in subsequent analysis. Social interaction was defined as the average relative distance from the divider and the percentage of time spent orienting at 45˚ to 90˚, and these parameters were measured and analyzed using our previously described computer vision software written in Python (available at https://github.com/stednitzs/daniopen). To account for changes in nutrition between fish, standard length was measured as previously described [51].

Optomotor response was assessed using a previously described "virtual reality" system for assessing zebrafish behavior, measuring swim response in 7 dpf larvae to concentric rings simulating motion toward the center of a container [99]. Briefly, we used infrared illumination to simultaneously record the swim responses of 9 AB × TU larvae at a time in 10 cm shallow glass

containers filled with EM. Larvae were imaged at 30 frames per second. Visual stimulus was projected on a screen underneath the dishes for 20 seconds and consisted of concentric rings moving toward the dish center, followed by a 20-second refractory period. Responses are the average of 46 to 59 stimulus trials per fish, presented over 1 hour.

## Sparse mosaic neuronal labeling

For sparse mosaic recombination of GFP and RFP transgenes in vTel[y321] neurons, *y321Et; UAS-bloswitch; hsp70lB3* larvae were heat shocked 24 hours ahead of terminal sampling (at 6 dpf or 13 dpf) by immersing sterile flasks in a 37˚C water bath for 30 minutes [53]. Larvae were returned to 28˚C for an additional day following heat shock.

## Immunocytochemistry

Larval zebrafish were immunolabeled as previously described [6]. Briefly, 7 dpf larvae were humanely killed with MS-222, fixed in 4% paraformaldehyde at room temperature overnight, permeabilized in phosphate-buffered saline (PBS) with 0.5% Triton X-100 (PBSTx), and then blocked overnight at room temperature in PBSTx with 5% normal goat serum, 2% bovine serum albumin, and 1% DMSO. Larvae were then treated with primary antibodies overnight at room temperature diluted in blocking solution at the concentrations indicated below, washed, and treated with secondary antibodies diluted 1:1,000 in PBSTx for 6 hours at room temperature. Finally, larvae were washed in PBSTx, eyes, lower jaws, and tails were removed, and the remaining tissue was mounted in Prolong Diamond anti-fade mountant (Invitrogen Cat# P36970). At 14 dpf, larvae were humanely killed on ice and prefixed in 4% paraformaldehyde for 1 hour. The midbrain and forebrain were dissected in PBS, removed, and fixed overnight at room temperature in 4% paraformaldehyde. We used a modified CUBIC protocol for clearing and immunolabeling dissected 14 dpf larval brains [100]. Brains were rinsed in PBS and incubated in CUBIC 1 solution (25% wt urea, 25% wt Quadrol, and 15% wt Triton X-100 in dH$_2$O) at 37˚C for 2 to 3 days. Brains were then washed, blocked, and incubated with primary antibodies as described above. After additional washing steps, brains were incubated with secondary antibodies diluted 1:100 in PBSTx overnight at room temperature. Brains were then briefly washed, incubated in CUBIC 2 solution (25% wt urea, 50% wt sucrose, and 10% wt triethanolamine in dH$_2$O) at room temperature for 6 hours, and mounted in Prolong Diamond anti-fade mountant (Invitrogen Cat# P36970). The following primary antibodies were used: mouse anti-GFP (1:100; Invitrogen Cat #A-11120) and rabbit anti-mCherry (1:100; Novus Biologicals Cat #2–25157). The following secondary antibodies were used: Alexa Fluor 488 goat anti-mouse IgG (Invitrogen Cat #A28175) and Alexa Fluor 546 goat anti-rabbit IgG (Invitrogen Cat #A-11035).

## Fluorescence in situ hybridization

The 7 dpf larvae were humanely killed with MS-222 and fixed in 4% paraformaldehyde for 3 hours at room temperature. Larvae were washed in PBS with 0.1% Tween-20 (PBST), dehydrated through a methanol series, and incubated in 100% methanol at −20˚C overnight. *c1qa* RNA was then detected using the Advanced Cell Diagnostics (ACD) RNAscope Multiplex Fluorescence V2 Kit (ACD #323100), adapting the manufacturer's protocols as previously described [101]. Briefly, larvae were air-dried and incubated with Protease Plus for 1 hour and 15 minutes at room temperature, washed in PBST, and incubated with prewarmed custom probes (*c1qa*, *c1qb*, *gad1b*, negative control) at 40˚C overnight. A custom probe against *gad1b* was used as positive control, and the ACD RNAscope 4-plex Negative Control Probe was used as negative control (ACD #321831), and we observed no difference in control probe signal

between conditions. Next, larvae were washed in saline sodium citrate (SSC) buffer with 0.1% Tween-20 (SSC/Tw), refixed in 4% paraformaldehyde for 10 minutes at room temperature, and hybridized with the appropriate amplifier DNA (AMP-1, AMP-2) for 30 minutes each at 40˚C. Each probe was then developed by sequential application of the appropriate HRP reagent (HRP-C1, HRP-C2) for 15 minutes each at 40˚C and fluorophore (Akoya Biosciences Opal 690 and Opal 520) for 30 minutes each at 40˚C. HRP blocker was applied between each channel for 15 minutes at 40˚C. Following probe amplification and labeling, larvae were immunolabeled with rabbit anti-Cherry primary antibody and Alexa Fluor 568 secondary antibody as described above but substituting PBST for PBSTx.

## Microscopy

For quantification of neuronal morphology, microglia infiltration, and *c1q* RNA localization, fixed and immunostained larval brains were imaged on a Leica TCS SP8 X (Leica Microsystems, Wetzlar, Germany) or Zeiss LSM 880 (Carl Zeiss Microscopy, LLC, Thornwood, New York, USA) confocal microscope. Neuronal arbors and microglia were imaged with a 40× water-immersion lens (1.10 NA). Z stacks were acquired at 1 μm per slice through the entire forebrain. To ensure comparable resolution across samples and conditions, projections outside of a single field of view at 40× were captured by tiling multiple z stacks in Leica LAS X 3.1.5.16308 software (Leica Microsystems, Wetzlar, Germany). *c1q* RNA localization was imaged with a 63× oil-immersion lens (1.40 NA) and 3× zoom. Z stacks were acquired at 0.3 μm per slice. Microglial dynamics were imaged live in 7 dpf *y321Et; UAS:GFP; mpeg1: mCherryTg* heterozygotes raised GF or CVZ on a Nikon CSU-W1 SoRa spinning disk confocal microscope (Nikon Instruments, Melville, New York, USA) with 20× lens, imaging a z stack with 1 μm slice depth encompassing the larval forebrain every 30 or 60 seconds for 20 minutes. Time series imaged every 30 seconds were downsampled to every 60 seconds for consistency across the dataset.

## Image analysis

Neuronal morphology was extracted from confocal z stacks by 3D segmentation in Imaris software (Oxford Instruments, Zurich, Switzerland) as previously described [53]. Briefly, Imaris Filament Tracer was used in "AutoPath" mode to semiautomatically segment neurites based on RFP fluorescence signal. The number of recombined cells in each brain varied from none to dozens; only arbors that could be accurately distinguished without overlap from neighboring cells were segmented. Statistics and a.swc representation were exported from each filament object for further analysis and visualization. The number of cells in the GFP-positive population was estimated by threshold-based surface creation using the "split touching objects" function and identical estimated cell size applied across all samples and conditions.

vTel$^{y321}$ neuropil density was estimated from confocal Z stacks using 3D surface objects created in Imaris. This analysis was performed blind to condition. For each image, a 3D surface corresponding to the surface of the forebrain was created by semiautomatic local contrast detection in a brightfield image. This surface was used to mask *vTel$^{y321}$:GFP* and *mpeg1: mCherry* channels, excluding signal outside of the forebrain. Signal-based intensity thresholding was used to create a 3D surface of the remaining *vTel$^{y321}$:GFP* signal. The volume of the *vTel$^{y321}$:GFP* surface was computed and divided by the volume of the forebrain surface to calculate the density of *vTel$^{y321}$* neuropil in the forebrain. Forebrain microglia were quantified by placing an Imaris spots object on the cell body of each microglia and extracting microglial position and number.

Microglial morphology was quantified by semiautomatic signal-based segmentation with Imaris Filament Tracer in each image and, for measurements of morphological variance, across each time series. Cumulative intensity projections were generated in the FIJI distribution of ImageJ [102], manually segmented to exclude *mpeg1:mCherry* signal from circulating macrophages outside of the brain, and % area filled was measured.

*c1qa* RNA localization was quantified by semiautomatic signal-based segmentation using the surfaces function in Imaris. Briefly, *mpeg1:mCherry* signal was used to mask each image, create a 3D region of interest for analysis of *c1qa* signal in each cell, and estimate microglial volume. Imaris Spots was used to detect and quantify *c1qa* puncta in each cell. Intensity-based thresholds were applied equally across all images and conditions.

## Image registration

Average CVZ and GF forebrains were generated separately using vTel[y321] GFP signal as a reference. Specifically, a single brain with representative size and orientation was first chosen as a reference for each condition. Each additional brain was then registered to these templates using the Computational Morphology Toolkit (CMTK; http://nitrc.org/projects/cmtk), executing the following parameters via the terminal: -awr 01 -T 4 -X 26 -C 8 -G 80 -R 4 -A '—accuracy 0.4' -W '—accuracy 0.4' -s. The resulting transformed vTel[y321] GFP images were then averaged to generate a single average forebrain for each condition. Each original image was then registered again, this time to the condition average in CMTK with the parameters described above. This generated transformed images and image rotation, translation, scaling, shearing, and centering coordinates used to achieve that transformation. These transformation coordinates were then applied to SWC-formatted neurons using Natverse package functions in R [103]. Formatting neurons as SWC files converts them into a matrix of (x,y,z) coordinates so that they can be read across platforms. For each condition, transformed neurons were exported from R in SWC format and imported into the average vTel[y321] GFP forebrain for 3D visualization in Imaris software (Oxford Instruments, Concord MA).

## Morpholino injection

One- to two-cell stage embryos were injected with 1 to 2 nl of 0.3 mM translation-blocking morpholino (GeneTools) targeting *irf8* (*irf8* MO[atg]; 5′-TCAGTCTGCGACCGCCCGAGTT-CAT-3′) [57]. Off-target effects were controlled by comparison to uninjected embryos and embryos injected with Random Control-25N morpholino mixture (GeneTools) injected at concentration equal to the experimental morpholino. Morpholinos were prepared as a 1-mM stock solution, which was diluted to working concentration and coinjected with 0.05% phenol red solution. MO-injected embryos included for analysis were morphologically normal and survived at rates comparable to embryos injected with random control morpholino (59% to 74% for Random Control-25N, 74% for *irf8* MO[atg]).

## Single-cell RNA sequencing analysis

Microglial expression profiles from CV and GF larvae were compared in RStudio [104] using scRNAseq data from Massaquoi and colleagues [62]. *mpeg1*[+] immune cell cluster 36 was subclustered using Seurat v4.0.4 [105] software package for R, v4.1.1 [106]. Briefly, cluster 36 reads were scaled and centered using the ScaleData command, and data were clustered using FindNeighbors and FindClusters commands. RunPCA and ElbowPlot were used to evaluate how many principal components to include. RunUMAP was applied using 4 principal components and resolution 1.25, which was empirically determined by evaluating separation of clusters with resolution values ranging from 0.5 to 2.0. Significantly enriched genes in each of the 9

resulting clusters were generated using the FindConservedClusters command. Expression differences between CV and GF cells in each subcluster were extracted using the FindMarkers command. Transcriptional similarity between microglia and macrophages makes it difficult to separate the two based on individual marker genes. To identify microglial clusters, we used a "microglial fingerprint" based on previously described microglial gene expression in zebrafish, mice, and humans [107]. We extracted genes previously identified as common between 5 dpf and 7 dpf zebrafish that are also conserved in murine and human microglia and added several additional well-studied microglial genes to this unbiased conserved fingerprint, including *apoeb*, *c1qa*, *c1qb*, *hexb*, *mafb*, *plxnb2a*, *sall1a*, and *slc7a7* (Fig 6B and S1 Table) [105]. It is likely that zebrafish microglia strongly express additional genes not represented on this list, but we hypothesized that focusing only on genes conserved in mice and humans would facilitate identification of cell clusters with features that are well studied across taxa.

### Statistics

Groups were statistically compared using Prism 8 software (Graphad, San Diego, California, USA) as described in the figure legends. Gaussian distribution of each group was examined by a D'Agostino–Pearson test of skewness and kurtosis. Unpaired *t* tests were applied to data with Gaussian distribution and equal standard deviation, and Welch's correction was applied if standard deviation of the 2 groups was unequal. Mann–Whitney *U* tests were applied to data that were not normally distributed. More than 2 groups were compared using one-way analysis of variance (ANOVA) with Tukey's multiple comparisons test, or if the data were not normally distributed, a Kruskal–Wallis test with Dunn's multiple comparisons test. $P < 0.05$ was considered statistically significant. Outliers were not removed from any experimental groups. SPSS Statistics 26 (IBM, New York, USA) was used for hierarchical clustering and cluster analysis based on 13 morphological features extracted from individual neurons in Imaris. Hierarchical clustering measured the squared Euclidean distance between neurons using between-groups linkage of measurements transformed by z-scores. Underlying morphological features were extracted by principal axis factoring using a varimax rotated component matrix for variable assignments and eigenvalue cutoff of 1. In both 7 dpf and 14 dpf datasets, 3 factors accounted for the majority of variance in the measurements (77.46% and 80.39%, respectively).

## Supporting information

**S1 Fig. Developmental size and social behavior.** (**A**) Standard length is reduced in 14 dpf XGF (aqua) larvae relative to CVZ (gray) siblings ($n$ = 57 CVZ and 66 GF larvae; unpaired *t* test). (**B**) Percent time oriented at 45–90˚ in XGF larvae and CVZ siblings, binned according to standard length. (**C**) Swim speed is not significantly different between XGF larvae and CVZ siblings ($n$ = 55 CVZ and 67 GF larvae; Mann–Whitney *U* test). ns, not significant; [**], $P < .01$. Solid red line represents the median; dotted red lines represent the upper and lower quartiles. Data underlying this figure are available on figshare: https://figshare.com/projects/Bruckner_et_al_Data/136756.
(TIF)

**S2 Fig. The microbiota refines vTel[y321] targeting.** Dorsal (top) and lateral (bottom) views of vTel[y321] neurons from (**A**) 7 dpf CVZ (gray) and GF (aqua) larvae and (**B**) 14 dpf CVZ (gray) and XGF (aqua) larvae registered to an average vTel[y321] nucleus (transparent 3D model) from each condition and developmental stage. Average vTel[y321] nuclei do not incorporate sparse neuronal somata at the periphery, which are within the forebrain boundary. Neurons right of

the dotted line in the factor analysis plots in Fig 3H and 3J are indicated in orange.
(TIF)

**S3 Fig. The microbiota is required for normal forebrain microglial Z position.** (**A**) Representative dorsal views of maximum-intensity projections of *mpeg1:mCherryTg* (microglia and macrophages, magenta) and vTel[y321] GFP (neurons, green) in 7 dpf CVZ (gray) or GF (aqua) larvae. Dotted lines indicate approximate forebrain boundary, segmented from the corresponding brightfield image. (**B-D**) The number of forebrain microglia, normalized to total forebrain volume (**B**), is reduced in GF larvae relative to CVZ siblings, while vTel[y321] neuropil density (**C**) and the position of the center of mass of the vTel[y321] neuropil, normalized to forebrain size (**D**), are increased in GF larvae relative to CVZ siblings ($n = 8$ CVZ and 8 GF larvae; unpaired $t$ test). (**E**) 3D position of individual forebrain microglia from CVZ (left) and GF (right) larvae ($n = 319$ microglia from 8 CVZ larvae 208 microglia from 8 GF larvae). Z position is indicated by color. (**F**) Average microglial Z position is significantly reduced in GF larvae relative to CVZ siblings ($n = 319$ microglia from 8 CVZ larvae 208 microglia from 8 GF larvae; Welch's $t$ test). *, $P < .05$; **, $P < .01$; ****, $P < .0001$. Solid red line represents the median; dotted red lines represent the upper and lower quartiles. Data underlying this figure are available on figshare: https://figshare.com/projects/Bruckner_et_al_Data/136756
(TIF)

**S4 Fig. The microbiota does not dramatically shift immune cell clustering.** (**A**) The distribution of cells within each *mpeg1.1*[+] Cluster 36 subcluster (left) is similar for cells from larvae raised CVZ (right, green) or GF (right, blue). (**B**) *c1qa* and *c1qb* expression is largely exclusive to Cluster 36 immune cells. Data underlying Fig 6A were also used to create (**A**) above, and with the data underlying (**B**), are available on figshare: https://figshare.com/projects/Bruckner_et_al_Data/136756.
(TIF)

**S1 Movie. Social orienting and place preference in a representative 14 dpf CVZ larva.**
(AVI)

**S2 Movie. Social orienting and place preference in a representative 14 dpf XGF larva.**
(AVI)

**S3 Movie. 3D rotation of vTel[y321] neurons from 7 dpf CVZ larvae, registered to an average vTel[y321] nucleus (transparent 3D model).** Neurons right of the dotted line in the factor analysis plots in Fig 4B and 4D are indicated in orange.
(MP4)

**S4 Movie. 3D rotation of vTel[y321] neurons from 7 dpf GF larvae, registered to an average vTel[y321] nucleus (transparent 3D model).** Neurons right of the dotted line in the factor analysis plots in Fig 4B and 4D are indicated in orange.
(MP4)

**S5 Movie. 3D rotation of vTel[y321] neurons from 14 dpf CVZ larvae, registered to an average vTel[y321] nucleus (transparent 3D model).** Neurons right of the dotted line in the factor analysis plots in Fig 4B and 4D are indicated in orange.
(MP4)

**S6 Movie. 3D rotation of vTel[y321] neurons from 14 dpf XGF larvae, registered to an average vTel[y321] nucleus (transparent 3D model).** Neurons right of the dotted line in the factor analysis plots in Fig 4B and 4D are indicated in orange.
(MP4)

**S7 Movie. CVZ forebrain *mpeg1:mCherryTg*-positive microglial dynamics.** Maximum-intensity Z-projection of a 20-minute spinning disc confocal time series in a representative CVZ larva, 1 volume imaged per minute.
(MP4)

**S8 Movie. GF forebrain *mpeg1:mCherryTg*-positive microglial dynamics.** Maximum-intensity Z-projection of a 20-minute spinning disc confocal time series in a representative GF larva, 1 volume imaged per minute.
(MP4)

**S1 Table. Differential expression of a microglial fingerprint across immune cell clusters.**
(XLSX)

**S2 Table. GO term enrichment in amoeboid (cluster 1) microglia.**
(XLSX)

**S3 Table. GO term enrichment in ramified (cluster 4) microglia.**
(XLSX)

**S4 Table. Genes differentially expressed between CV and GF conditions in amoeboid (cluster 1) microglia.**
(XLSX)

**S5 Table. Genes differentially expressed between CV and GF conditions in ramified (cluster 4) microglia.**
(XLSX)

## Acknowledgments

We thank members of the Eisen and Washbourne laboratories for feedback on earlier versions of the manuscript, members of the Guillemin laboratory and University of Oregon Microbial Ecology and Theory of Animals (META) Center for gnotobiology support, Adam Christensen and the University of Oregon Zebrafish Facility Staff for animal husbandry, and Harold Burgess for the generous gift of zebrafish lines.

## Author Contributions

**Conceptualization:** Joseph J. Bruckner, Sarah J. Stednitz, Philip Washbourne, Judith S. Eisen.

**Formal analysis:** Joseph J. Bruckner, Sarah J. Stednitz, Max Z. Grice, Dana Zaidan.

**Funding acquisition:** Joseph J. Bruckner, Sarah J. Stednitz, Karen Guillemin, Philip Washbourne, Judith S. Eisen.

**Investigation:** Joseph J. Bruckner, Sarah J. Stednitz, Max Z. Grice, Dana Zaidan.

**Resources:** Michelle S. Massaquoi, Johannes Larsch, Alexandra Tallafuss, Karen Guillemin.

**Visualization:** Joseph J. Bruckner.

**Writing – original draft:** Joseph J. Bruckner.

**Writing – review & editing:** Joseph J. Bruckner, Sarah J. Stednitz, Max Z. Grice, Dana Zaidan, Michelle S. Massaquoi, Johannes Larsch, Alexandra Tallafuss, Karen Guillemin, Philip Washbourne, Judith S. Eisen.

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
