## [Editor Report · Decision Letter 0]

5 Apr 2022

Dear Dr Eisen, 

Thank you for submitting your manuscript entitled "The microbiota promotes social behavior by modulating microglial remodeling of forebrain neurons." for consideration as a Research Article by PLOS Biology.

Your manuscript has now been evaluated by the PLOS Biology editorial staff, as well as by an academic editor with relevant expertise, and I am writing to let you know that we would like to send your submission out for external peer review.

Once your full submission is complete, your paper will undergo a series of checks in preparation for peer review. Once your manuscript has passed the checks it will be sent out for review. To provide the metadata for your submission, please Login to Editorial Manager (https://www.editorialmanager.com/pbiology) within two working days, i.e. by Apr 07 2022 11:59PM.

If your manuscript has been previously reviewed at another journal, PLOS Biology is willing to work with those reviews in order to avoid re-starting the process. Submission of the previous reviews is entirely optional and our ability to use them effectively will depend on the willingness of the previous journal to confirm the content of the reports and share the reviewer identities. Please note that we reserve the right to invite additional reviewers if we consider that additional/independent reviewers are needed, although we aim to avoid this as far as possible. In our experience, working with previous reviews does save time. 

If you would like to send previous reviewer reports to us, please email me at kdickson@plos.org to let me know, including the name of the previous journal and the manuscript ID the study was given, as well as attaching a point-by-point response to reviewers that details how you have or plan to address the reviewers' concerns. 

Given the disruptions resulting from the ongoing COVID-19 pandemic, please expect some delays in the editorial process. We apologise in advance for any inconvenience caused and will do our best to minimize impact as far as possible.

Kind regards,

Kris

Kris Dickson

Neurosciences Senior Editor/Section Manager

PLOS Biology

kdickson@plos.org

---

## [Decision Letter · Decision Letter 1]

12 May 2022

Dear Dr Eisen,

Thank you for your patience while your manuscript "The microbiota promotes social behavior by modulating microglial remodeling of forebrain neurons." was peer-reviewed at PLOS Biology. It has now been evaluated by the PLOS Biology editors, an Academic Editor with relevant expertise, and by three independent reviewers. 

In light of the reviews, which you will find at the end of this email, we would like to invite you to revise the work.

As you will see, all three reviewers feel that the topic is interesting and that the study has the potential to provide a broad impact. However, they also raised a number of concerns and feel that a fair bit of additional functional data is required to bolster your mechanistic conclusions. In particular, they've pointed out the need for more data to specifically relate the microbiota effects on microglial and neuron phenotypes to each other, analyses to temporally relate the cellular changes with the social changes to bolster causality claims, and data to strengthen the role for c1q in this process. We ask that you thoroughly address these and other concerns that the reviewers' reports raised during revision.

Given the extent of revision needed, we cannot make a decision about publication until we have seen the revised manuscript and your response to the reviewers' comments. Your revised manuscript is likely to be sent for further evaluation by all or a subset of the reviewers.

**IMPORTANT - SUBMITTING YOUR REVISION**

*Re-submission Checklist*

*Published Peer Review*

*PLOS Data Policy*

*Blot and Gel Data Policy*

Sincerely,

Kris

Kris Dickson, Ph.D (she/her)

Neurosciences Senior Editor/Section Manager

PLOS Biology

kdickson@plos.org

REVIEWS:

Reviewer's Responses to Questions

PLOS authors have the option to publish the peer review history of their article (what does this mean?). If published, this will include your full peer review and any attached files.

Reviewer #1: No

Reviewer #2: No

Reviewer #3: No

Reviewer #1: Bruckner et al.'s manuscript explores the mechanism by which the microbiota is able to modulate the development of behavioral traits, using the zebrafish as a model. They established a robust protocol allowing to raise gut bacteria-free embryos and larvae and showed in previous publications that microbiota is important for development of some specific innate behaviors. They also showed that a specific population of neurons in the ventral telencephalon was required for these behaviors. Here they test the hypothesis that the microbiota may be required for normal development of these telencephalic neurons.

They indeed observe that embryos raised without bacteria until 7dpf (GF/XGF) show abnormalities in these neurons. They show that the vTel population has increased neurite arborization compared to control. They also show that microglia in GF larvae is reduced in number in the telencephalon. They therefore test the reasonable hypothesis that lack of microbiota reduces forebrain microglia population, in turn leading to increased vTel neurite arborization. The findings are of great interest but the manuscript lacks clarity in places and requires more experiments to strengthen the conclusions made. The three main weaknesses are: i) the absence of evidence for telencephalic specificity in increase of neurite arborization and decreased microglia ii) the reduced amount of data directly supporting causality from low number of microglia to high amount of neurite branching, and iii) the very preliminary functional data suggesting c1q complement involvement in the reduced interaction between neurons and microglia. 

Specific concerns:

- Tracing of the vTel in Figure S2 strongly suggest that the nucleus (and maybe the whole telencephalon) is smaller in the GF animals at both 7 and 14dpf. The authors need to measure total telencephalic volume to assess whether the forebrain is smaller in GF (only overall larval size was measured) and if so, whether the size difference is also present in other brain areas.

- The increased arborization observed in vTel is compelling and extremely interesting. Is increased branching also observed in other neuronal populations? Or is it specific to vTel? The same question needs to be assessed for microglia. Is drop in microglia number telencephalon-specific or seen throughout the brain (author can normalize to tectal area to test forebrain specificity as tectal microglia is easy to image and quantify)? 

- In the GF (Fig. 4A,B) and microglia morphants (Fig. 4E), the number of vTel neurons looks increased on the panels shown. Quantification of neuron numbers need to be provided for these two experiments in addition to neuropil density.

- Differentiating microglia are present in the midbrain and forebrain as early as 3dpf. To further strengthen the causal link from less microglia to increased neurite branching, the authors should FACS GFP+ microglia from 4dpf and 5dpf control and inject them into GF larval forebrain at the same stage and assess whether injection rescues GF, reducing vTel branching and larval behavior.

- The scRNAseq analysis in Fig 6 is done from clustering RNAseq from whole larvae. It is therefore not surprising to see subtle difference in the signature of gene expression in the microglia clusters as some of these cells would be localized in and/or around the gut and experiencing a very different surveying landscape. Not sure whether the proteosomal and lysosomal difference identified is meaningful for the telencephalic microglia specifically. Unless the authors can better articulate the meaning of the difference in GO terms for the telencephalon population specifically and validate it, I would remove this part of the manuscript.

- I am a bit worried about the fluorescent in situ shown on Fig 7. There is a substantial amount of signal outside of the cells in which the gene is expressed. How do the authors recognize signal vs background? 

- The c1q MO experiment is affecting c1q from the start of its expression everywhere it is expressed. The fact that there is an increase in neuropil in the telencephalon of the MO injected could be very indirect, for instance by affecting the gut of the embryo, which in turn would lead to telencephalic phenotype. If the authors want to address this, they should FACS mpeg:mCherry cells from c1q crispants (MO embryos do not deal with FACS well), inject them into forebrain of normal 3dpf vTelGFP larvae and image neuropil at 7dpf. 

Minor point: 

- As there is not difference in microglia morphology or kinetics, Fig 5 should be moved to suppl. Data

Reviewer #2: Bruckner et al. report interesting evidence that the microbiota controls social behavior in zebrafish by influencing the localization and function of microglia, which in turn regulate the arbors of forebrain neurons that modulate social behavior.

Key findings include that social behavior at 14 dpf is altered in animals raised in germ-free conditions for 0-7 dpf, and that arbors of forebrain neurons controlling this behavior are more dense in these animals. Microglia are somewhat less abundant in the forebrain of GF animals. The abnormalities in the forebrain of GF animals can be at least partially rescued by diverse, individual bacterial taxa. Morphants with reduced microglia have increased forebrain neuropil, apparently providing the first evidence that microglia prune neuronal arbors in zebrafish. Analysis of scRNA-seq data led the authors to examine c1q expression and function; they obtain evidence that c1q expression is reduced in microglia from GF animals, and that c1q knockdown increases forebrain neuropil.

This is a very interesting paper, because it provides a link between the microbiota, social behavior, and microglia-neuron interactions. Some of the effect sizes are small, but overall the study is convincing. Below are some points that the authors should address, with additional experiments and/or revisions to the text.

What is the critical period for the effect of microbiota on social behavior? i.e., does restoration of microbiota at 6 dpf, 5 dpf, … rescue subsequent social behavior? The social behavior assays are useful at stages well past when the authors examined. Do the social abnormalities in XGF animals persist to adulthood (or at least past 14 dpf)?

It would strengthen the case for causality if the authors can link the timing of changes in microglia and neuronal arbors to the timing of social deficits discussed above. In other words, does the microbiota critical period for social behavior coincide with changes in the arbors or localization of microglia?

Another other experiment that would strengthen the microbiota-microglia-arbor linkage proposed by the authors would be to examine social behavior in irf8 mutants (or other mutants lacking microglia). Similarly, are microglia required for mono-associated microbes to rescue neuropil density (i.e. repeat experiment in Figure 4D in mutants lacking microglia)?

When is the reduction in forebrain microglia first apparent in GF animals? Are microglia also reduced in other areas of the CNS?

Relative to the original irf8 morpholino paper (Li et al. 2011), the knockdown reported by the authors seems very week. Why do so many microglia (apparently more than half) remain in the irf8 morphants? 

The c1q morphant experiment is not as convincing as the other experiments reported in the paper. What is the knockdown efficiency? The increase in neuropil density seems small. Is it possible to perform a rescue experiment in which c1q is restored to the morphants? Are there social deficits in c1q morphants? Analysis of a c1q mutant would greatly strengthen the conclusions of this section of the paper.

Reviewer #3: This is a fascinating study adding further evidence for the role of the microbiome in social behaviour

1. Which microbiota was used for conventionalizing? Was there a difference between CV and CVZ controls?

2. Was hyperactivity observed in GF larvae as has been reported previously (eg Phelps et al 2017) ?

3. Was the body length (and hence development state) normalized for arborization and subsequent experiments?

4. Was the microglia distribution evaluated at 14 dpf XGF and CVZ as was done for arbor analysis?

5. Microglial abundance and neuropil density was ascertained at what age in experiments with morpholino against the microglial gene irf8?

6. Was social behaviour assayed upon administration of different bacterial strains?

Minor

The term dysbiosis is heavily disputed in the field and should be avoided

The authors articulate and justify very well the concept of using zebrafish as a model of social behaviour but could also point out how there is increasing emphasis on its role as an excellent tool to dissect mechanisms in microbiota-gut-brain axis function (eg PMID: 34653349)

---

## [Decision Letter · Decision Letter 2]

26 Aug 2022

Dear Dr Eisen,

Thank you for your patience while we considered your revised manuscript "The microbiota promotes social behavior by modulating microglial remodeling of forebrain neurons." for publication as a Research Article at PLOS Biology. This revised version of your manuscript has been evaluated by the PLOS Biology editors, the Academic Editor, and the original reviewers.

Based on the reviews and our Academic Editor's assessment of your revision, we are likely to accept this manuscript for publication. Given the concerns raised by Reviewer 2, in order to move forward with this study at PLOS Biology, we ask that you remove the functional c1q data from the study (i.e. Fig 7E-I) and rewrite the work accordingly to remove discussion of these points throughout the manuscript and instead simply discussing the potential implications of these gene changes. 

When making these changes, please also make sure to address the data and other policy-related requests detailed at the bottom of this email. In particular, please note our requirements for data deposition and ensure that the deposited summary data is clearly linked to individual figures in your study, and that each figure legend directs readers to the specific summary data for the individual figure panels. At the moment, the scRNAseq data is not provided in a manner clearly linked to the specific figures that were generated for the study, and the FigShare link that was provided did not work.

As you address these items, please also review your reference list to ensure that it is complete and correct, particularly given the removal of data that we have requested. If you have cited papers that have been retracted, please include the rationale for doing so in the manuscript text, or remove these references and replace them with relevant current references. Any changes to the reference list should be mentioned in the cover letter that accompanies your revised manuscript.

We expect to receive your revised manuscript within two weeks. 

*Published Peer Review History*

*Press*

Sincerely,

Kris

Kris Dickson, Ph.D. (she/her)

Neurosciences Senior Editor/Section Manager,

kdickson@plos.org,

PLOS Biology

DATA POLICY:

Note that we do NOT require all raw data. Rather, we need data provided that reflect the individual quantitative observations/graphs/heatmaps/etc that underlie the data summarized in the figures and results of your paper. These summary datasets can be made available in one of the following forms:

1) Supplementary files (e.g., excel). Please ensure that all data files are uploaded as 'Supporting Information' and that they are invariably referred within the manuscript, the figure legends, and the Description field when uploading your files). For these supplementary files use the following format verbatim: S1 Data, S2 Data, etc. Multiple panels of a single or even several figures can be included as multiple sheets in one excel file that is saved using exactly the following convention: S1_Data.xlsx (using an underscore) and with each sheet clearly labeled as to the Figure panel it refers to.

2) Deposition in a publicly available repository. Please also provide the accession code or a reviewer link so that we may view your data before publication. Deposition online also requires that the repository links are invariably referred within the manuscript, the figure legends, and the Description field when uploading your files). For these files use the following format verbatim: S1 Data, S2 Data, etc. Multiple panels of a single or even several figures can be included as multiple sheets in one excel file that is saved using exactly the following convention: S1_Data.xlsx (using an underscore) and with each sheet clearly labeled as to the Figure panel it refers to.

Regardless of the method selected, we will need you to provide the individual numerical values that underlie the summary data displayed in each of the following figure panels as they are essential for readers to assess your analysis and to reproduce it:

Fig1D,F,H,I; Fig2B-E; Fig3A-F;H;J; Fig4C,D,F,H,I,J; Fig5B,C,E,F; Fig6A,B; Fig7B,C,D (rest of Fig7 will be removed).

SuppFig1A-C; SuppFig3B-F; SuppFig4A

***DO NOT FORGET: Please also ensure that each figure legend in your manuscript includes a specific statement directing readers on where the underlying data can be found, and ensure your supplemental data file/s also has such legends.***

Please also ensure that your Data Statement in the submission system accurately describes where your data can be found.

DATA NOT SHOWN?

- Please note that per journal policy, we do not allow the mention of "data not shown", "personal communication", "manuscript in preparation" or other references to data that is not publicly available or contained within this manuscript. If such statements are present, please either remove mention of these data or provide figures presenting the results and the data underlying the figure(s).

Reviewer remarks:

Reviewer #1: The authors have responded satisfactorily to my queries. In my opinion, it doesn't need further revision 

Reviewer #2: I appreciate the authors' efforts to revise the manuscript, but the additional analysis of c1q morphants does not provide evidence that c1q is required for social behavior. In the c1q morphants, there is a 21% increase in neurite density without any measurable effect on social behavior. The limitations of MO, including variable, transient knockdown and the possibility of off-target effects, remain a major concern with the c1q studies. The function of c1q should be substantiated through analysis of mutants, or these data should be removed from the paper.

Reviewer #3 (John Cryan): The authors have addressed all our concerns and the paper will make a great addition to the field.

---

## [Editor Report · Decision Letter 3]

19 Sep 2022

Dear Dr Eisen,

Thank you for the submission of your revised Research Article "The microbiota promotes social behavior by modulating microglial remodeling of forebrain neurons." for publication in PLOS Biology. On behalf of my colleagues and the Academic Editor, Kelly Monk, I am pleased to say that we can in principle accept your manuscript for publication, provided you address any remaining formatting and reporting issues. These will be detailed in an email you should receive within 2-3 business days from our colleagues in the journal operations team; no action is required from you until then. Please note that we will not be able to formally accept your manuscript and schedule it for publication until you have completed any requested changes. Please note that we will also need you to confirm that the scRNAseq data has been deposited in GEO and to provide us with a DOI. This needs to be available before we can publish your work.

Please also take a minute to log into Editorial Manager at http://www.editorialmanager.com/pbiology/, click the "Update My Information" link at the top of the page, and update your user information to ensure an efficient production process.

PRESS

Sincerely, 

Kris

Kris Dickson, Ph.D. (she/her)

Neurosciences Senior Editor/Section Manager

PLOS Biology

kdickson@plos.org